

# Chemical cycling and deposition of atmospheric mercury
# in Polar Regions: review of recent measurements and
# comparison with models
**Hélène Angot[1], Ashu Dastoor[2], Francesco De Simone[3], Katarina Gårdfeldt[4],**
**Christian N. Gencarelli[3], Ian M. Hedgecock[3], Sarka Langer[5], Olivier Magand[6, 1],**
**Michelle N. Mastromonaco[4], Claus Nordstrøm[7], Katrine A. Pfaffhuber[8], Nicola**
**Pirrone[9], Andrei Ryjkov[2], Noelle E. Selin[10, 11], Henrik Skov[7], Shaojie Song[10],**
**Francesca Sprovieri[3], Alexandra Steffen[12], Kenjiro Toyota[12], Oleg Travnikov[13],**
**Xin Yang[14], Aurélien Dommergue[1, 6]**
[1]Univ. Grenoble Alpes, Laboratoire de Glaciologie et Géophysique de l'Environnement
(LGGE), 38041 Grenoble, France
[2]Air Quality Research Division, Environment and Climate Change Canada, 2121
TransCanada Highway, Dorval, Quebec H9P 1J3, Canada
[3]CNR-Institute of Atmospheric Pollution Research, Division of Rende, UNICAL-
Polifunzionale, 87036 Rende, Italy
[4]Department of Chemistry and Chemical Engineering, Chalmers University of Technology
SE-412 96 Göteborg, Sweden
[5]IVL Swedish Environmental Research Institute, P.O. Box 530 21, 400 14 Göteborg, Sweden
[6]CNRS, Laboratoire de Glaciologie et Géophysique de l'Environnement (LGGE), 38041
Grenoble, France
[7]National Environmental Research Institute, Frederiksborgvej 399, 4000 Roskilde, Denmark
[8]Norwegian Institute for Air Research (NILU), P.O. Box 100, 2027 Kjeller, Norway
[9]CNR-Institute of Atmospheric Pollution Research, Area della Ricerca di Roma 1,
Monterotondo, 00015 Rome, Italy
[10]Department of Earth, Atmospheric and Planetary Sciences, Massachusetts Institute of
Technology, Cambridge, MA, USA





[11]Institute for Data, Systems, and Society, Massachusetts Institute of Technology, Cambridge,
MA, USA
[12]Air Quality Research Division, Environment and Climate Change Canada, Toronto, Ontario
M3H 5T4, Canada
[13]Meteorological Synthesizing Centre, East of EMEP, 2nd Roshchinsky proezd, 8/5, 115419
Moscow, Russia
[14]British Antarctic Survey, Cambridge, United Kingdom
Correspondence to: A. Dommergue (aurelien.dommergue@univ-grenoble-alpes.fr)

## 37    **Abstract**

Mercury (Hg) is a worldwide contaminant that can cause adverse health effects to wildlife and
humans. While atmospheric modeling traces the link from emissions to deposition of Hg onto
environmental surfaces, large uncertainties arise from our incomplete understanding of
atmospheric processes (oxidation pathways, deposition, and reemission). Atmospheric Hg
reactivity is exacerbated in high latitudes and there is still much to be learned from Polar
Regions in terms of atmospheric processes. This paper provides a synthesis of the
atmospheric Hg monitoring data available in recent years (2011-2015) in the Arctic and in
Antarctica along with a comparison of these observations with numerical simulations using
four cutting-edge global models. The cycle of atmospheric Hg in the Arctic and in Antarctica
presents both similarities and differences. Coastal sites in the two regions are both influenced
by springtime atmospheric Hg depletion events and by summertime snowpack reemission and
oceanic evasion of Hg. The cycle of atmospheric Hg differs between the two regions
primarily because of their different geography. While Arctic sites are significantly influenced
by Northern Hemispheric Hg emissions especially in winter, coastal Antarctic sites are
significantly influenced by the reactivity observed on the East Antarctic ice sheet due to
katabatic winds. Based on the comparison of multi-model simulations with observations, this
paper discusses whether the processes that affect atmospheric Hg seasonality and inter-annual
variability are appropriately represented in the models, and identifies research gaps in our
understanding of the atmospheric Hg cycling in high latitudes.



## 1 Introduction

Mercury (Hg) can be emitted to the atmosphere by natural geological sources (e.g., volcanic emissions) and a variety of anthropogenic activities (e.g., coal combustion, artisanal and small-scale gold mining) (UNEP, 2013b). The dominant form of atmospheric mercury is gaseous elemental mercury (Hg(0)) (Lindberg and Stratton, 1998). Hg(0) has an atmospheric lifetime of 0.5 to 1 year (Selin, 2009) and can therefore be transported worldwide. It can be oxidized into highly reactive and water soluble gaseous and particulate divalent species (Hg(II) and Hg(p), respectively) that can deposit onto environmental surfaces (e.g., land, surface oceans) through wet and dry processes (Lindqvist and Rodhe, 1985). Upon deposition, mercury can be reemitted to the atmosphere or converted – in aquatic systems – to methylmercury (Driscoll et al., 2013). Anthropogenic activities have altered the global geochemical cycle of mercury, enhancing the amount of mercury circulating in the atmosphere and surface oceans by at least a factor of three (Lamborg et al., 2014; Amos et al., 2015).

Methylmercury is a worldwide contaminant of seafood that can cause adverse effects on the developing nervous system of vulnerable populations (AMAP, 2015). The Minamata Convention on mercury – global treaty to protect human health and the environment from mercury – was opened for signature in October 2013 (UNEP, 2013a). To date, the Convention has been signed by 128 countries and ratified by 28. It will enter into force once it is ratified by 50 nations. As noted in the preamble of the Convention, Arctic ecosystems and indigenous communities are particularly vulnerable due to the biomagnification of mercury and contamination of traditional foods. In order to reduce mercury effects, the pathway from emissions to human and environmental impacts needs to be traced. Atmospheric modeling provides a first step by tracing the link from emissions to deposition onto environmental surfaces. Deposition of mercury in a particular region depends on the magnitude and speciation of domestic and foreign emissions, and on the oxidative capacity of the atmosphere that transforms Hg(0) to deposited divalent species (UNEP, 2015). Deposition is partly offset by the revolatilization of a fraction of deposited mercury. Large uncertainties associated with the models arise as a result of our incomplete understanding of atmospheric processes (e.g., oxidation pathways, deposition, and reemission) (Kwon and Selin, 2016). Atmospheric mercury reactivity is exacerbated in high latitudes and there is still much to be learned from Polar Regions in terms of atmospheric processes.



First discovered in 1995 (Schroeder et al., 1998), Atmospheric Mercury Depletion Events
(AMDEs) are observed in springtime throughout the Arctic (Lindberg et al., 2001; Berg et
al., 2003a; Poissant and Pilote, 2003; Skov et al., 2004; Steffen et al., 2005) as a result of
the oxidation of Hg(0) by reactive bromine species (Lu et al., 2001; Brooks et al., 2006;
Sommar et al., 2007). AMDEs can lead to a deposition of ~ 100 tons of mercury per year to
the Arctic (Ariya et al., 2004; Skov et al., 2004; Dastoor et al., 2015). The fate of mercury
deposited onto the snowpack during AMDEs is still a matter of debate in the scientific
mercury community (Steffen et al., 2008). Several studies reported significant reemission
(e.g., Ferrari et al., 2005; Brooks et al., 2006; Kirk et al., 2006; Sommar et al., 2007;
Dommergue et al., 2010a) although a fraction of mercury may likely accumulate within the
snowpack (Hirdman et al., 2009; Larose et al., 2010). While the Arctic has been extensively
monitored – with hundreds of publications focusing on AMDEs, measurements are sporadic
in Antarctica. To the best of the author's knowledge, only eleven studies dealing with
atmospheric mercury in Antarctica (and using modern instrument) have been published
(Ebinghaus et al., 2002; Sprovieri et al., 2002; Temme et al., 2003; Brooks et al., 2008a;
Brooks et al., 2008b; Dommergue et al., 2012; Pfaffhuber et al., 2012; Angot et al., 2016a;
Angot et al., 2016b; Nerentorp Mastromonaco et al., 2016; Wang et al., 2016). The earliest
studies showed the occurrence of AMDEs in coastal Antarctica after polar sunrise. The latest
studies highlighted new atmospheric processes in the Antarctic boundary layer – both in
winter and summertime – leading to the formation and subsequent deposition of reactive
mercury. In the meantime, several studies showed that the Antarctic Plateau plays a key role
in influencing the cycle of atmospheric mercury at a continental scale.
The first objective of this paper is to provide a synthesis of the atmospheric mercury
monitoring data available in recent years (2011-2015) in Polar Regions. Secondly, we provide
a comparison of these observations with numerical simulations of atmospheric mercury
concentrations using cutting-edge global models. Finally, this paper identifies research gaps in
our understanding and modeling of the atmospheric mercury cycling in high latitudes.





## 2   Experimental Section

### 2.1   Measurements of atmospheric mercury species

#### 2.1.1   Definitions

Hg(0), Hg(II), and Hg(p) are the most abundant mercury species in the atmosphere. Atmospheric Hg(0) is easily and accurately measured in Polar Regions (Steffen et al., 2008; Dommergue et al., 2010b). Hg(p) and Reactive Gaseous Mercury (RGM) – the latter consisting of various gaseous Hg(II) compounds – are operationally defined. Total Gaseous Mercury (TGM) refers to the sum of Hg(0) and Hg(II), and Reactive Mercury (RM) to the sum of RGM and Hg(p).

#### 2.1.2   Instrumentation

Measurements of atmospheric mercury species were performed at various sites in the Arctic and in Antarctica over the 2011-2015 period (Fig. 1). All Hg(0) measurements reported in this paper were performed using a Tekran gas phase analyzer (Model 2537), and all RGM and Hg(p) measurements using a Tekran speciation unit (1130/1135) (Table 1). The Tekran 2537 analyzer is based on the amalgamation of mercury onto a gold cartridge followed by a thermal desorption and detection by an integrated cold vapor atomic fluorescence spectrometer (CVAFS) at 253.7 nm (Fitzgerald and Gill, 1979; Bloom and Fitzgerald, 1988). The analysis of Hg(0) is semi-continuous and the presence of two gold cartridges allows alternating sampling and desorption modes. At all sampling sites, the sample air stream was prefiltered either through a Tekran speciation unit or through a sodalime trap and/or a PTFE (polytetrafluoroethylene) filter (Table 1). Some researchers report ambient air collected at Polar sites as TGM (Ebinghaus et al., 2002), instead of Hg(0), but the PTFE filter on the front of the analyzer inlet most likely removes RGM and thus only Hg(0) is collected and analyzed (Steffen et al., 2002; Steffen et al., 2008). Due to the extremely cold and dry air in Antarctica, no heated sampling line was used and no sodalime was applied at TR, DC, and DDU. Collected at 5 to 15 min intervals at the various sites, Hg(0) measurements are reported here as hourly averages. RGM and Hg(p) measurements at ALT and ANT were performed using a Tekran speciation unit – connected to a 2537 analyzer through a PTFE heated sampling line – through a multistep procedure as described elsewhere (Lindberg et al., 2002) using an impactor inlet (2.5 µm cut-off aerodynamic diameter at 10 L/min), a KCl-coated quartz



annular denuder in the 1130 unit, and a quartz regenerable particulate filter (RPF) in the 1135
unit.

**Quality assurance and quality control procedures**

Auto-calibrations of the 2537 analyzers were performed every 25 to 72 hours at the various
sites using an internal mercury permeation source. The accuracy of this permeation source
was checked at least once per year against manual injections using a Tekran 2505 mercury
vapor calibration unit and following a strict procedure adapted from Dumarey et al. (1985).
The detection limit for Hg(0) measurements is 0.10 ng/m$^3$ according to the instrument manual
(Tekran, 2011). Based on experimental evidence, the average systematic uncertainty for
Hg(0) measurements is of ~ 10 % (Slemr et al., 2015). There is no robust calibration
technique of the Tekran speciation unit and no certified reference material available. There is
growing evidence that RGM and Hg(p) might suffer from significant biases and interferences
(Lyman et al., 2010; Gustin et al., 2013; Jaffe et al., 2014; Huang et al., 2013; Kos et al.,
2013), and that RGM concentrations might be underestimated by as much as a factor of 2 - 13
(Gustin et al., 2016). Despite these limitations, the Tekran speciation unit is currently the best
available automated method, and Hg(p) and RGM measurements can be used as first
estimates to evaluate atmospheric models. Maintenance operations on the Tekran
2537/1130/1135 instruments and screening criteria for data validation/invalidation were
performed according to the directives of the standard operational procedure (SOP) from
CAMNet (Canadian Mercury Measurement Network), AMNet (United States Atmospheric
Mercury Network), or GMOS (Global Mercury Observation System) (Steffen et al., 2012;
D'Amore et al., 2015).

## 2.2 Global mercury simulations

The current study is based on multi-model simulations performed as part of the Mercury
Modeling Task Force (MMTF) under the GMOS project (Travnikov et al., in preparation).
Four global models (ECHMERIT, GEM-MACH-Hg, GEOS-Chem, and GLEMOS) were
applied for evaluating monthly-averaged atmospheric mercury concentrations and deposition
at various Arctic and Antarctic ground-based sites for the year 2013. Additionally, GEM-
MACH-Hg and GEOS-Chem provided hourly-averaged data from 2011 to 2014 to allow
investigations of inter-annual variability. A brief description of the parameterization of the
four models is given below. The models differ significantly in their description of mercury





atmospheric chemistry and their parameterization of processes specific to Polar Regions (i.e.,
AMDEs, oceanic evasion, and re-emissions from the snowpack).
**2.2.1 ECHMERIT**
ECHMERIT is a fully-coupled model, based on the Atmospheric General Circulation Model
(AGCM) ECHAM5, and a mercury chemistry module, developed at the Institute for
Atmospheric Pollution of the National Research Council (CNR-IIA) of Italy (Jung et al.,
2009; De Simone et al., 2014; De Simone et al., 2016). The base mechanism includes
oxidation of Hg(0) by OH and $O_3$ in the gas and aqueous (in-cloud) phases (reactions R1 to
R3). Rate constants of reactions (R1) to (R3) are from Sommar et al. (2001), Hall (1995), and
Munthe (1992), respectively.
$Hg(0) + OH \rightarrow Hg(II)$           **(R1)**
$Hg(0) + O_3 \rightarrow Hg(II)$           **(R2)**
$Hg(0)_{(aq)} + O_{3\,(aq)} \rightarrow Hg(II)_{(aq)}$      **(R3)**
Oxidant fields (OH/$O_3$) are imported from MOZART (Model for Ozone and Related
Chemical Tracers) (Emmons et al., 2010). In the base run used for this work bromine
chemistry is not included, and there is no parameterization of AMDEs. ECHMERIT
implements dynamically calculated ocean emissions for all ice-free basins, including Polar
Regions, as described in De Simone et al. (2014), and a prompt re-emission of 60 % of
deposited mercury over ice (Selin et al., 2008).
**2.2.2 GEM-MACH-Hg**
GEM-MACH-Hg is a mercury version of the Environment and Climate Change Canada's
(ECCC's) current operational air quality forecast model – Global Environmental Multi-scale –
Modelling air quality and Chemistry (GEM-MACH). GEM-MACH-Hg is an on-line model,
meaning that the meteorology is simulated in-step with the chemistry, and includes
representation of physicochemical processes of mercury based on the ECCC's previous
mercury model – GRAHM (Dastoor and Larocque, 2004; Dastoor et al., 2008; Durnford et
al., 2010; Durnford et al., 2012; Kos et al., 2013; Dastoor et al., 2015). The horizontal
resolution of the model for this study is $1° \times 1°$ latitude/longitude. Hg(0) is oxidized in the
atmosphere by OH (R1) and bromine (reactions (R4) to (R6), X = Br or BrO). The rate
constant of (R1) is from Sommar et al. (2001), but scaled down by a coefficient of 0.34 to
take into account possible dissociation reactions (Tossell, 2003; Goodsite et al., 2004). Rate



constants of reactions (R4) to (R6) are from Donohoue et al. (2006), Dibble et al. (2012) and
Goodsite et al. (2004), respectively. Aqueous-phase reduction reactions are not included.
$Hg(0) + Br \rightarrow Hg(I)Br$ **(R4)**
$Hg(I)Br \rightarrow Hg(0) + Br$ **(R5)**
$Hg(I)Br + X \rightarrow Hg(II)X$ **(R6)**
OH fields are from MOZART (Emmons et al., 2010) while BrO is derived from 2007-2009
satellite observations of BrO vertical columns. The associated Br concentration is then
calculated from photochemical steady state according to equation (1), where $J_{BrO}$ is the BrO
photolysis frequency, and $k_1 = 2.1 \ 10^{-11} \ cm^3 \ molecule^{-1} \ s^{-1}$ and $k_2 = 1.2 \ 10^{-12} \ cm^3 \ molecule^{-1} \ s^{-1}$
$^1$ are the rate coefficients for the $BrO + NO \rightarrow Br + NO_2$ and $Br + O_3 \rightarrow BrO + O_2$
reactions, respectively (Platt and Janssen, 1995).
$$\frac{[Br]}{[BrO]} = \frac{J_{BrO} + k_1 \ [NO]}{k_2 \ [O_3]} \qquad (1)$$
Durnford et al. (2012) developed and implemented a dynamic multilayer snowpack/meltwater
parameterization allowing the representation of deposition and reemission of mercury.
Oceanic evasion of Hg(0) is activated if there is open water and the temperature at the air-sea
interface is -4 °C or greater (Dastoor and Durnford, 2014). In addition, Hg(0) released from
sea ice melting is also taken into account. The parameterization of AMDEs is based on Br
production and chemistry, and snow reemission of Hg(0) (Dastoor et al., 2008).
**2.2.3 GEOS-Chem**
GEOS-Chem (v9-02) is a global chemical transport model driven by assimilated
meteorological data from the NASA GMAO Goddard Earth Observing System (Bey et al.,
2001). It couples a 3-D atmosphere (Holmes et al., 2010), a 2-D mixed layer slab ocean
(Soerensen et al., 2010), and a 2-D terrestrial reservoir (Selin et al., 2008) with a horizontal
resolution of 2° × 2.5° latitude/longitude. Three mercury tracers (Hg(0), Hg(II), and Hg(p))
are tracked in the atmosphere (Amos et al., 2012). Mercury fluxes at terrestrial and ocean
surfaces are described in Song et al. (2015). A two-step oxidation mechanism initialized by Br
atoms is used (reactions (R4) to (R6), X = Br or OH). Br fields are archived from a full-
chemistry GEOS-Chem simulation (Parrella et al., 2012) while rate constants of reactions
(R4) to (R6) are from Donohoue et al. (2006), Balabanov et al. (2005), and Goodsite et al.
(2012), respectively. Some model setups related to Polar Regions are implemented in v9-02 of



the model as described in details in Holmes et al. (2010). 5 pptv of BrO – at the low end of
concentrations reported by Neuman et al. (2010) – is added in the springtime Arctic
(Antarctic) boundary layer during March-May (August-October) over areas with sea ice,
sunlight, stable conditions, and temperatures below 268 K. The associated Br concentration is
then calculated from photochemical steady state according to equation (1) assuming that $O_3$ is
depleted to 2 ppbv. Additionally, a snowpack reservoir is added. It accumulates deposited
mercury and releases it as Hg(0) under sunlit conditions in a temperature-dependent way.

### 2.2.4  GLEMOS

GLEMOS (Global EMEP Multi-media Modelling System) is a multi-scale chemical transport
model developed for the simulation of environmental dispersion and cycling of different
chemicals including mercury (Travnikov and Ilyin, 2009). The model simulates atmospheric
transport, chemical transformations and deposition of three mercury species (Hg(0), Hg(II),
and Hg(p)). The atmospheric transport of tracers is driven by meteorological fields generated
by the Weather Research and Forecast (WRF) modelling system (Skamarock et al., 2007) fed
by the operational analysis data from ECMWF. The model in the base configuration has a
horizontal resolution of $1° \times 1°$. The base mechanism includes oxidation of Hg(0) by OH
(R1) and $O_3$ (R2) in the atmosphere. Rate constants are from Sommar et al. (2001) and Hall
(1995), respectively. The model also includes in-cloud oxidation of Hg(0) by OH, $O_3$, and Cl
with associated rate constants from Gårdfeldt et al. (2001), Munthe (1992), and Lin and
Pehkonen (1999), respectively. In-cloud reduction by $SO_3^{2-}$ is also implemented, with an
associated rate constant from Petersen et al. (1998). Reactant fields are imported from
MOZART (Emmons et al., 2010).
The parameterization of AMDEs in Polar Regions is based on Br chemistry following the
two-step mechanism (R4)-(R6) described in Holmes et al. (2010). Br concentrations are
extracted from p-TOMCAT (parallel-Tropospheric Off-Line Model of Chemistry and
Transport) results (Yang et al., 2005). GLEMOS includes an empirical parameterization of
prompt-reemission from snow. It is assumed that reemission occurs only from newly
deposited mercury in the presence of solar radiation. Two competing processes are
considered: photoreduction and ageing of deposited mercury with the characteristic times of 1
day and 10 days, respectively. It is also assumed that all reduced mercury is immediately
reemitted back to the atmosphere. The aged fraction of mercury does not undergo reduction
and is accumulated within the snowpack. No mercury evasion from the ocean is implemented.



### 2.3 Goodness-of-fit statistics between modeled and observed data

The Nash-Sutcliffe efficiency (NSE, Nash and Sutcliffe, 1970) indicates how well the plot of observed versus simulated data fits the 1:1 line – NSE = 1 corresponding to the perfect match. NSE is defined as one minus the sum of the absolute squared differences between the simulated and observed values normalized by the variance of the observed values:

$$NSE = 1 - \frac{\sum_{i=1}^{N}(O_i - S_i)^2}{\sum_{i=1}^{N}(O_i - \bar{O})^2} \qquad (2)$$

The root mean square error (RMSE) gives the standard deviation of the model prediction error (in the same units of simulated and observed values). A smaller value indicates better model performance. It is calculated as follows:

$$RMSE = \sqrt{\frac{1}{N} \sum_{i=1}^{N}(S_i - O_i)^2} \qquad (3)$$

The percent bias (PBIAS, in %) measures the average tendency of the simulated values to be larger or smaller than their observed ones. The optimal value of PBIAS is 0. PBIAS is calculated as follows:

$$PBIAS = 100 \frac{\sum_{i=1}^{N}(S_i - O_i)}{\sum_{i=1}^{N} O_i} \qquad (4)$$

NSE, RMSE, and PBIAS were calculated by using the R package "hydroGOF" (Zambrano-Bigiarini, 2014).

### 3 Results and Discussion

#### 3.1 Arctic sites

##### 3.1.1 Observations

Fig. 2a shows monthly box plots of all data collected at the four Arctic sites. The average Hg(0) value in the Arctic over the 2011-2014 period is $1.46 \pm 0.33$ ng m$^{-3}$. This concentration falls within the range of what is observed in the Northern Hemisphere (Sprovieri et al., this issue-a). The highest mean is at AND ($1.55 \pm 0.15$ ng m$^{-3}$ over the 2011-2015 period), which is closer from European industrialized areas than other sites and experiences less frequent and pronounced AMDEs in spring (see section 3.1.1.2). There is a clear Hg(0) concentration gradient (except from June to August): AND > NYA > SND > ALT.





The Hg(0) concentration data from the four Arctic sites for the period 2011-2015 are
presented as monthly box and whisker plots in Fig. 3. Information regarding annually- and
monthly-based statistics at the three sites can be found in Tables 2 and 3, respectively. The
annual medians at NYA and AND (Table 2) suggest a low inter annual variability in the
distribution of Hg(0) concentrations. Conversely, there is a high degree of inter-annual
variability at ALT and SND driven by the intensity of spring and summertime processes. This
will be addressed in the following sections.
The mean seasonal variation of Hg(0) concentrations at Arctic sites is displayed in Fig. 4a.
Summer refers to June – August, fall to September – November, winter to December –
February, and spring to March – May. Hg(0) concentrations exhibit a strong and consistent
seasonal pattern year after year, as already reported by others (Steffen et al., 2005;  Berg et
al., 2013). Hg(0) concentrations reach a distinct maximum in summer at ALT, SND, and
NYA (mean concentrations of $1.63 \pm 0.37$, $1.63 \pm 0.37$, and $1.60 \pm 0.23$ ng m$^{-3}$, respectively).
In late summer the concentrations start to decrease and reach in fall a mean value of $1.28 \pm$
$0.12$ ng m$^{-3}$ at ALT, $1.36 \pm 0.11$ ng m$^{-3}$ at SND, and $1.46 \pm 0.16$ ng m$^{-3}$ at NYA. In winter,
concentrations increase slightly and are significantly higher than in fall at the three sites ($p$
value $< 0.0001$ at the three sites, Mann-Whitney test). Springtime reflects the lowest Hg(0)
concentrations with mean values of $1.11 \pm 0.58$ ng m$^{-3}$ at ALT, $1.28 \pm 0.51$ ng m$^{-3}$ at SND,
and $1.38 \pm 0.38$ ng m$^{-3}$ at NYA. The seasonal cycle is more pronounced at ALT than at SND
and NYA. Hg(0) concentrations at AND exhibit an opposite seasonal cycle with a
significantly ($p$ value $< 0.0001$, Mann-Whitney test) higher mean concentration in winter
($1.67 \pm 0.11$ ng m$^{-3}$) than in summer ($1.48 \pm 0.12$ ng m$^{-3}$), in line with the seasonality reported
at Pallas, Finland (67°22'N, 26°39'E) (Berg et al., 2001;  Sprovieri et al., this issue-a). The
mechanisms which cause the seasonal variation of Hg(0) concentrations at Arctic sites are
discussed in the following sections.
**3.1.1.1 Wintertime advection of Hg from mid-latitudes**
Several studies highlighted that the Arctic is significantly influenced by atmospheric pollution
from mid-latitudes – phenomenon known as Arctic haze – during wintertime (Barrie et al.,
1981;  Heintzenberg et al., 1981;  Shaw, 1982;  Heidam et al., 1999;  Heidam et al., 2004;
Bourgeois and Bey, 2011;  Nguyen et al., 2013). Dastoor and Larocque (2004) used an on-line
model to explain the observed seasonal variations in atmospheric mercury circulation and
showed frequent episodes of mercury transport from mid-latitudes sources to the Arctic in
winter. Similarly, Hirdman et al. (2009) attributed the highest 10 % of all wintertime Hg(0)



data at NYA to transport of air masses especially from Europe. Higher Hg(0) concentrations
in winter compared to fall at ALT, SND, and NYA can therefore be attributed to the
meteorological differences in the seasonal circulation patterns (Dastoor and Larocque, 2004).
Higher concentrations in winter at AND compared to the three other Arctic sites can be
attributed to the powerful advection of air masses from Europe at this site (Durnford et al.,

338    2010).

**3.1.1.2 Springtime AMDEs**

AMDEs in the Arctic are defined as Hg(0) concentrations below 1.00 ng m$^{-3}$ (Steffen et al.,
2005; Cobbett et al., 2007). Based on this threshold, AMDEs occur in 39 %, 28%, 15%, and
1% of the 2011-2014 springtime observations at ALT, SND, NYA, and AND, respectively.
The fact that ALT experiences stronger and more frequent AMDEs than other Arctic sites
could be due to air masses circulation patterns. Several studies indicated that a large fraction
of the AMDEs reported at NYA and AND are suspected to result from the long-range
transport of air masses containing depleted Hg(0) from areas over the Arctic Ocean (Gauchard
et al., 2005; Sommar et al., 2007; Berg et al., 2008; Steen et al., 2011; Berg et al., 2013). A
statistical analysis on the results from a Lagrangian particle dispersion model (FLEXPART)
and Hg(0) concentrations measured at NYA was performed by Hirdman et al. (2009) to
identify source regions of high- and low-Hg air masses. The authors concluded that the lowest
10% of the Hg(0) data at NYA in spring were strongly associated with transport across the
sea-ice covered Arctic Ocean at low altitudes – areas where elevated BrO concentrations are
seen in the atmospheric column by satellite observations (e.g., Lindberg et al., 2002).
Similarly, a correlation of AMDEs with wind direction at ALT supports the origin of
depletion events over the Arctic Ocean (Cole and Steffen, 2010). The less frequent and
pronounced AMDEs at AND may be explained by the fact that this site is farther away from
the source areas of AMDEs (Berg et al., 2008).
Over the 2011-2015 period, AMDEs at NYA are evenly distributed between April and May as
38 and 38% respectively, and fewer in March and June (14 and 10 % of the time,
respectively). This result is in good agreement with the distribution reported by Berg et al.
(2013) over the 2000-2009 period. Conversely, AMDEs are more frequent in April (41 %)
than in May (32 %) at ALT, while less frequent in April (34 %) than in May (43 %) at SND.
Interestingly, the analysis of the ALT dataset from 1995 to 2007 by Cole and Steffen (2010)
revealed that, over time, the month of maximum AMDE activity shifted from May to April.
On the contrary, the analysis of the NYA dataset from 2000 to 2009 by Berg et al. (2013) did



not evidence such a change in the timing frequency of AMDEs. The reason for this shift in
timing of AMDEs at ALT is not fully understood but could be due to local meteorology (Cole
and Steffen, 2010). The authors found that the length, magnitude, and frequency of AMDEs
decreased with increasing local temperature. These results are consistent with earlier studies
on the temperature dependence of the halogen chemistry initiating AMDEs and ozone
depletion events (Koop et al., 2000; Adams et al., 2002; Tarasick and Bottenheim, 2002;
Sander et al., 2006) and with a modeling study reporting that increasing surface air
temperature decreases the frequency of AMDEs (Chen et al., 2015) . However, considering
the fact that AMDEs observed at Arctic sites often result from the transport of depleted air
masses, local temperature might not be the key explanatory parameter. Moore et al. (2014)
showed that AMDEs and ozone depletion events near Barrow, Alaska, are directly linked to
sea-ice dynamics. According to the authors, depletion events are favored by consolidated sea-
ice cover but both Hg(0) and $O_3$ concentrations immediately recover to near-background
concentrations when air masses cross open leads within a day before measurements. The
authors attributed this recovery of concentrations to changes in boundary-layer dynamics
induced by sea-ice leads, causing significant convective mixing with non-depleted air masses
aloft. Further work is needed to establish the degree to which sea-ice dynamics across the
Arctic might influence the inter-annual variability of AMDEs at the various Arctic sites.
Indeed, AMDEs occurred at ALT in 36 % (2011), 51 % (2012), 50 % (2013), and 21 %
(2014) of the springtime observations, at SND in 37 % (2011), 16 % (2012), 36 % (2013), and
19 % (2014) of the springtime observations, and finally at NYA in 18 % (2011), 13 % (2012),
16 % (2013), 20 % (2014), and 6 % (2015) of the springtime observations.
Several studies reported RGM and Hg(p) concentrations during AMDEs at Arctic sites
(Lindberg et al., 2002; Berg et al., 2003a; Steffen et al., 2003; Aspmo et al., 2005;
Gauchard et al., 2005; Sprovieri et al., 2005a; Steen et al., 2011; Wang, 2015). Fig. 5 shows
box plots of the monthly concentrations of RGM and Hg(p) at ALT over the 2011-2014
period. A distinct annual cycle is highlighted in this figure. Hg(p) concentrations increase
from November through February likely due to the Arctic haze (Steffen et al., 2014), reach a
maximum in March and April due to AMDEs, and then decrease. RGM concentrations peak
in spring and then gradually decrease. The production of RGM in June and July – after the
AMDEs season – is observed every year and remains unexplained (Steffen et al., 2014).
While Hg(p) is the dominant species in early spring, a clear shift is observed, from the
predominance of Hg(p) to RGM in AMDEs occurring toward the end of spring. This shift has



already been evidenced at Churchill, Manitoba (Kirk et al., 2006), ALT (Cobbett et al., 2007),
and NYA (Steen et al., 2011), and has been shown to repeat year after year at ALT (Steffen et
al., 2014). Steffen et al. (2014) suggested that this shift is due to temperature and particle
availability. Using a detailed air-snowpack model for interactions of bromine, ozone and
mercury in the springtime Arctic, Toyota et al. (2014) proposed that Hg(p) is mainly produced
as $HgBr_4^{2-}$ through uptake of RGM into bromine-enriched aerosols after ozone is significantly
depleted in the air mass. In addition, Toyota et al. (2014) provided the temperature
dependence of these reactions which needs to be verified experimentally. Based on ten years
of data, Steffen et al. (2014) also reported higher levels of mercury in the snow when the
atmospheric conditions favored the formation of RGM. This springtime shift from the
predominance of Hg(p) to RGM in AMDEs likely directly impacts the amount of mercury
deposited onto the snowpack. This will be further discussed in section 3.1.2.2.
**3.1.1.3 Summer enhancement of Hg(0) concentrations**
According to Dastoor and Larocque (2004), advection of mercury from mid-latitudes to the
Arctic is insignificant in summer due to weak airflow movements and to a confined polar
front. The increase of Hg(0) concentrations in summer could be due to the reemission of
mercury deposited during springtime AMDEs. However, the comparison of the magnitude of
the springtime depletion and the magnitude of the summer enhancement at ALT suggests
otherwise. Mean springtime Hg(0) concentrations are lower – suggesting more intense and/or
frequent AMDEs – in 2012 ($0.97 \pm 0.53$ ng m$^{-3}$) and 2013 ($0.89 \pm 0.57$ ng m$^{-3}$) than in 2011
($1.19 \pm 0.59$ ng m$^{-3}$) and 2014 ($1.37 \pm 0.50$ ng m$^{-3}$), while mean summertime concentrations
are higher – suggesting more reemission – in 2011 ($1.81 \pm 0.37$ ng m$^{-3}$) and 2014 ($1.63 \pm 0.31$
ng m$^{-3}$) than in 2012 ($1.43 \pm 0.27$ ng m$^{-3}$) and 2013 ($1.65 \pm 0.41$ ng m$^{-3}$). Therefore, the
summer enhancement of Hg(0) concentrations is generally attributed to emissions from snow
and ice surfaces (Poulain et al., 2004;  Sprovieri et al., 2005b;  Sprovieri et al., 2005a;
Sprovieri et al., 2010;  Douglas et al., 2012) and/or to evasion from the ice-free surface waters
of the Arctic Ocean (Aspmo et al., 2006;  Andersson et al., 2008;  Hirdman et al., 2009;
Fisher et al., 2013;  Dastoor and Durnford, 2014;  Yu et al., 2014;  Soerensen et al., 2016).
The atmospheric mercury model (GRAHM) used by Dastoor and Durnford (2014) simulated
a first peak in Hg(0) concentrations driven by revolatilization from snowpack/meltwaters,
followed by a second peak driven by oceanic evasion – the timing of the peaks varying with
location and year. Additional modeling studies suggested that some of the mercury in surface
ocean waters may come from riverine input (Fisher et al., 2012;  Soerensen et al., 2016).





As can be seen in Fig. 3, Hg(0) concentrations are significantly higher ($p$ value < 0.0001,
Mann-Whitney test) during summer 2011 at ALT ($1.81 \pm 0.37$ ng m$^{-3}$) than during the
following summers ($1.57 \pm 0.35$ ng m$^{-3}$ in average). At SND, Hg(0) concentrations peak in
summer 2013 ($1.91 \pm 0.37$ ng m$^{-3}$ *vs.* $1.52 \pm 0.26$ ng m$^{-3}$ in average during summers 2011,
2012, and 2014). One possible explanation for this inter-annual variability is sea ice extent.
Daily  sea  ice  maps  can  be  obtained  from  http://www.iup.uni-
bremen.de/iuppage/psa/2001/amsrop.html (Spreen et al., 2008). ALT and SND are both
surrounded by multi-year ice. During summer 2011, the Hall Basin – waterway between
Greenland and Canada's northernmost island where ALT is located – was ice-free. During
summer 2013, sea ice extent was particularly low in the Greenland Sea – between Greenland
and the Svalbard archipelago. These large areas of ice-free surface waters might have led to
enhanced oceanic evasion near ALT, and SND in 2011 and 2013, respectively. Indeed, Yu et
al. (2014) reported a negative correlation between TGM and salinity over an Arctic ice-
covered region, suggesting that ice melting would enhance TGM concentrations. This
hypothesis  is  further  supported  by  wind  data  obtainable  from
http://climate.weather.gc.ca/historical_data/search_historic_data_e.html  and
http://villumresearchstation.dk/data/. At ALT, the summertime dominant wind direction is
from north-east but with frequent and strong winds from south/south-west (Hall Basin), in
line with results reported by Bilello (1973) and Cobbett et al. (2007). At SND, the dominant
wind direction is from south-west but the direction becomes more variable in summer with
winds also occurring from south and east (Bilello, 1973;  Nguyen et al., 2013). Yet a
comprehensive and systematic analysis of air masses back-trajectories and sea-ice extent is
required to further investigate parameters responsible for the observed inter-annual variability.
NYA is normally surrounded by open water in the summer. Therefore, oceanic emissions are
expected to act as a significant local source to NYA, while being a regional and diffuse source
at ALT and SND (Cole et al., 2013). However, the summer enhancement of Hg(0)
concentrations is weaker at NYA than at ALT and SND (Fig. 4a). The western coast of
Spitsbergen island, where NYA is located, was ice-free year-round over the period of interest
possibly preventing the build-up of mercury-enriched ice-covered surface waters in winter
and intense evasion in summer. Additionally, a comparative study was carried out at NYA
with measurements at both 12 m a.s.l. and 474 m a.s.l.. While Aspmo et al. (2005) found no
significant difference between Hg(0) concentrations at the two elevations, several studies
(Berg et al., 2003b;  Sprovieri et al., 2005b;  Sommar et al., 2007) reported that Hg(0)





concentrations at 12 m a.s.l. were higher in magnitude and exhibited a higher variability than
at 474 m a.s.l.. Evidence of volatile mercury evasion from snow and water surfaces was also
obtained, suggesting a cycling of mercury near the surface. Zeppelin station at 474 m a.s.l. is
typically positioned over or at the top of the marine boundary layer of the fjord valley
(Sommar et al., 2007) likely, at least partly, explaining why the summer enhancement of
Hg(0) concentrations is weaker at NYA.
In contrast to observations at ALT, SND, and NYA, Hg(0) concentrations reach a minimum
in summer at AND. Transport of air masses from Europe is dominant at AND (Durnford et
al., 2010) and could mask any variability induced by oceanic evasion. The mean Hg(0)
concentration in summer at AND (1.48 ± 0.12 ng m$^{-3}$ over the 2011-2015 period) is consistent
with the value of ~ 1.42 ng m$^{-3}$ reported at Pallas, Finland over the 2013-2014 period
(Sprovieri et al., this issue-a).

### 3.1.2  Comparison with models

Table 4 displays goodness-of-fit statistics between monthly-averaged modeled and observed
data in 2013. Except at ALT, modeled Hg(0) concentrations are biased-low suggesting that
the four global models tend to underestimate sources of Hg(0). The ability of the four models
to reproduce the observed seasonality of Hg(0) concentrations at Arctic sites in 2013 is shown
in Fig. 6a and discussed in the following sections. As mentioned in section 2.2, GEM-MACH-
Hg and GEOS-Chem provided hourly-averaged data from 2011 to 2014. The inter-annual
variability of the monthly Hg(0) concentration distribution at Arctic sites as simulated by the
two models is displayed in Fig. 7a while Table 5 shows the percent bias between hourly-
averaged modeled and observed data on a seasonal basis from 2011 to 2014.

**3.1.2.1 Seasonal variation**

**a) Winter**

All the models (except ECHMERIT) overestimate Hg(0) concentrations at ALT in January
and February 2013, but reproduce well the average value in December 2013 (Fig. 6a). It is
worth noting that the observed mean value in January/February 2013 (1.24 ± 0.13 ng m$^{-3}$) is
lower than the value observed in December 2013 (1.45 ± 0.07 ng m$^{-3}$) and lower than the
hemispheric background (1.30 – 1.60 ng m$^{-3}$ according to Sprovieri et al. (this issue-a)).
Additionally, the observed mean value in January/February 2013 is at the low end of values
reported at this period of the year at ALT from 2011 to 2014 (Fig. 3, 1.40 ± 0.16 ng m$^{-3}$ in
2011, 1.32 ± 0.09 ng m$^{-3}$ in 2012, and 1.47 ± 0.12 ng m$^{-3}$ in 2014). The inter-annual





variability of observed Hg(0) concentrations at ALT is not captured by models. Modeled
Hg(0) concentrations in January/February range from 1.48 ± 0.03 in 2014 to 1.54 ± 0.03 ng
m$^{-3}$ in 2011 and 2012 with GEOS-C hem and from 1.54 ± 0.06 in 2012 to 1.58 ± 0.04 ng m$^{-3}$
in 2013 with GEM-MACH-Hg. Similarly, the inter-annual variability of modeled Hg(0)
concentrations is low at other Arctic sites (Fig. 7a). The wintertime inter-annual variability of
observed Hg(0) concentrations might be driven by meteorology and mercury emissions in
mid-latitudes. However, the AMAP/UNEP (2010) global inventory of mercury anthropogenic
emissions (annual mean emission fields) was used for all simulated years (2011-2014) in both
GEOS-Chem and GEM-MACH-Hg, preventing the consideration of inter-annual changes in
anthropogenic emissions.
**b) Spring**
Springtime reflects the lowest Hg(0) concentrations at ALT, SND, and NYA due to the
occurrence of AMDEs (see section 3.1.1.2). This minimum is well reproduced by GEM-
MACH-Hg, GEOS-Chem, and GLEMOS at all three stations, but not reproduced by
ECHMERIT (Fig. 6a). It should be noted that there is no parameterization of AMDEs in the
latter. Interestingly, GLEMOS predicts a similar springtime minimum at AND in
contradiction with the seasonal pattern observed at this station (see section 3.1.1.2). This
discrepancy can likely be attributed to uncertainties in Br fields extracted from p-TOMCAT.
As discussed in section 3.1.1.2, AMDEs were less frequent at ALT in 2014. This lower
occurrence frequency is fairly well reproduced by GEM-MACH-Hg (61 % (2011), 43 %
(2012), 53 % (2013), and 36 % (2014)), but not at all by GEOS-Chem (4 % (2011), 6 %
(2012), 13 % (2013), and 37 % (2014)). A temperature-dependence of BrO concentrations is
implemented in GEM-MACH-Hg and Br$_2$ is assumed to occur only over consolidated sea-ice
which would change with changing meteorological conditions. Conversely, a constant value
of 5 pptv of BrO is added in the springtime Arctic boundary layer into GEOS-Chem v9-02.
However, updates to Arctic mercury processes will be implemented in v11-01 based on
Fisher et al. (2012) and Fisher et al. (2013) (http://wiki.seas.harvard.edu/geos-
chem/index.php/Mercury#Updates_to_Arctic_Hg_processes). BrO concentrations will
depend on temperature according to a relationship chosen to optimize spring Hg(0)
concentrations and the shift of peak depletion at ALT from May to April (see section 3.1.1.2).
It should also be noted that GEOS-Chem relies on GEOS-5 and GEOS-FP meteorological
fields in 2011-2013 and 2014, respectively. Simulations in Polar Regions can be very
sensitive to subtle changes in meteorological fields, especially during the AMDEs season,



which could at least partly explain the inter-annual variability of modeled AMDEs occurrence
frequencies.
Based on the work by Moore et al. (2014) showing the impact of sea-ice leads on AMDEs
(AMDEs might be favored by consolidated sea-ice cover, see section 3.1.1.2), real-time
distribution of sea-ice dynamics including presence of leads is needed. Contrarily to
conclusions by Moore et al. (2014), a recent modeling study (Chen et al., 2015) carried out
using GEOS-Chem v9-02 – but including an ice/snow module and riverine inputs as described
by Fisher et al. (2012) and Fisher et al. (2013) – showed that increasing sea ice lead
occurrence increases the frequency of AMDEs. These contradictory results highlight the fact
that further work is needed regarding the degree to which sea-ice dynamics across the Arctic
alters mercury chemistry in spring.
**c) Summer**
All the models (except ECHMERIT in which polar processes are not implemented) capture,
to some extent, the summertime Hg(0) enhancement. GLEMOS clearly underestimates
summertime mean concentrations at ALT and SND (Fig. 6a). This can be attributed to
missing reemissions and/or oceanic evasion. As mentioned is section 3.1.1.3, Dastoor and
Durnford (2014) suggested two distinct summertime maxima: a first one supported by
revolatilization from snowpack/meltwaters occurring from the end of May to mid-June at
ALT, and in June at NYA; a second one supported by oceanic evasion from mid-July to early
August at ALT and NYA. GEOS-Chem gives a summer maximum in June instead of July at
ALT, SND, and NYA. This time-lag might result from to the fact that oceanic evasion from
the Arctic Ocean is not implemented in v9-02. v11-01 of the model will include, among other
updates, new present-day (2009) fields for net primary productivity (NPP) based on Jin et al.
(2012), a UV-B dependence for Hg(II) reduction in seawater based on results of O'Driscoll et
al. (2006), updated Hg(0) emissions from snow, and a source of mercury from the snowpack
to the Arctic Ocean at the onset of snowmelt. In order for the models to reproduce the inter-
annual variability of Hg(0) concentrations, real-time distribution of areas of ice-free surface
waters along with the type of surface (ice/snow/snow-free bedrock) are needed.
**3.1.2.2 Reactive Mercury and deposition**
Year 2013 modeled monthly-averaged RM concentrations and wet/dry deposition are
displayed in Fig. 8a. GEOS-Chem, GEM-MACH-Hg, and GLEMOS predict increased RM
concentrations in spring, during the AMDEs season, consistent with the observed pattern at



ALT (Fig. 5) and NYA (Wang, 2015). The fact that ECHMERIT does not capture the spring enhancement is not surprising since the model does not implement any chemistry specific to Polar Regions. GLEMOS also predicts a RM spring maximum at AND, in line with the modeled Hg(0) spring minimum at this site (Fig. 6a). As discussed in section 3.1.2.1.b, this can likely be attributed to uncertainties in Br fields extracted from p-TOMCAT. Long-term measurements of RM in the Arctic are scarce and limited to ALT and NYA (data not presented here). According to Fig. 8a, all four models underestimate RM concentrations at ALT from at least January to April 2013. Similarly, the comparison of modeled RM concentrations at NYA with annual averages reported by Steen et al. (2011) and Wang (2015) suggest an underestimation of the concentrations by GEOS-Chem, GEM-MACH-Hg, and ECHMERIT.

According to the models, deposition of mercury peaks in spring at ALT and SND, consistent with the RM spring maximum. The deposition of mercury during AMDEs depends on temperature, relative humidity and aerosol contribution (Cobbett et al., 2007), and is higher when the atmospheric conditions favor the formation of RGM over Hg(p) (see section 3.1.1.2). Therefore, as suggested by Steffen et al. (2015), prevailing atmospheric conditions must be fully characterized in order to accurately evaluate the deposition of mercury. GEOS-Chem and GLEMOS both predict higher dry deposition in spring at NYA. Wet deposition is largely driven by precipitation – RM being readily scavenged by rain or snow, whereas dry deposition depends on the boundary layer stability and the type of the underlying surface (Cadle, 1991). Deposition of mercury in the Arctic is typically inferred from concentrations of total mercury in the snowpack (e.g., Steffen et al., 2014) or from a Hg(0) flux gradient method (Steffen et al., 2002; Brooks et al., 2006; Cobbett et al., 2007; Steen et al., 2009), and not through direct measurement of wet and dry deposition, making it difficult to evaluate the accuracy of models predictions. To the best of our knowledge, NYA is the only site out of the four Arctic sites where wet deposition measurements have been reported (Sprovieri et al., this issue-b). From May to December 2013, the observed net wet deposition flux is equal to 0.9 µg m$^{-2}$ while modeled fluxes amount to 1.7, 3.2, 2.8, and 2.4 µg m$^{-2}$ according to GLEMOS, GEOS-Chem, GEM-MACH-Hg, and ECHMERIT, respectively. All four models overestimate the wet deposition flux. Interestingly, all four models also overestimate the amount of precipitation (by a factor of 2.0, 2.2, 2.1, and 1.1, respectively. Data not shown). Several studies showed that the form of precipitation (rain *vs.* snow) influences the collection efficiency of the sampler. Lynch et al. (2003) and Prestbo and Gay (2009) found that the





annual collection efficiency is 89 % and 87.1 ± 6.5 %, respectively, at cold weather sites of
the United States and Canada experiencing snowfall in winter *vs* 98.8 ± 4.3 % at warm
weather sites (Prestbo and Gay, 2009). Assuming an annual 89 % collection efficiency of
snow at NYA does not narrow the gap between observed and modeled amounts of
precipitation. However, an annual 89 % collection efficiency at NYA seems generous
considering that snow falls year round and that strong wind (> 10 m s$^{-1}$) and blowing snow are
frequent, especially in winter (Maturilli et al., 2013).

### 602    3.2   Antarctic sites

### 603    3.2.1   Observations

Fig. 2b shows monthly box plots of all data collected in Antarctica (ground-based sites and
cruises). Hg(0) concentrations from the ANT cruises displayed in Fig.2b refer to data
collected when R/V Polarstern operated within the marginal sea ice region (8 July – 23 July
2013, 25 July – 9 August 2013, 28 August – 5 October 2013) (Nerentorp Mastromonaco et
al., 2016). Similarly, Hg(0) concentrations from the OSO cruise refer to data collected at
latitude > 60°S. Hg(0) concentrations measured during the ANT and OSO cruises are
somewhat higher than values at ground-based Antarctic sites. The average value at Antarctic
sites is 0.96 ± 0.32 ng m$^{-3}$, i.e. 35% lower than the average value at Arctic sites (see section
3.1). This result is consistent with the North-to-South Hg(0) decreasing gradient reported by
Sprovieri et al. (this issue-a), and with values reported at Southern Hemisphere mid-latitudes
sites (Angot et al., 2014;  Slemr et al., 2015).
The Hg(0) concentration data from the three Antarctic ground-based sites for the period 2011-
2015 are presented as monthly box and whisker plots in Fig. 9. Information regarding
annually- and monthly-based statistics at the three sites can be found in Tables 2 and 3,
respectively. The annual medians for 2011-2015 at TR and 2012-2015 at DDU (Table 2)
suggest a low inter annual variability in the distribution of Hg(0) concentrations. Conversely,
Hg(0) concentrations are notably higher in 2015 than in 2012 and 2013 at DC. This trend is
more apparent from Fig. 9b, especially from March to September. It is worth noting that in
2015 measurements were performed at a different location within the "clean area" (the
instrument was moved from one shelter to another). Additionally, following the January 2014
instrument failure, a new Tekran instrument operated in 2015. The combination of these two
elements likely, at least partly, explains the offset observed in 2015. Despite this offset, the
seasonal trends of Hg(0) repeat from year to year at DC (see below).





The mean seasonal variation of Hg(0) concentrations at Antarctic ground-based sites is
displayed in Fig. 4b. Summer refers to November – February, fall to March – April, winter to
May – August, and spring to September – October. At TR, the Hg(0) concentrations are
significantly ($p$ value $< 0.0001$, Mann-Whitney test) higher in winter ($0.98 \pm 0.06$ ng m$^{-3}$) than
in summer ($0.89 \pm 0.29$ ng m$^{-3}$), in good agreement with the seasonal variation reported at TR
by Pfaffhuber et al. (2012) from February 2007 to June 2011, and at Neumayer (NM) by
Ebinghaus et al. (2002). Contrarily, Hg(0) concentrations at DDU are slightly but significantly
($p$ value $< 0.0001$, Mann-Whitney test) higher in summer ($0.88 \pm 0.32$ ng m$^{-3}$) than in winter
($0.84 \pm 0.11$ ng m$^{-3}$). On the high-altitude Antarctic plateau at DC, Hg(0) concentrations
exhibit a distinct maximum in fall ($1.45 \pm 0.27$ ng m$^{-3}$) and a minimum in summer ($0.78 \pm$
$0.46$ ng m$^{-3}$). The mechanisms which cause the seasonal variation of Hg(0) concentrations at
Antarctic sites are discussed in the following sections.

### 639 3.2.1.1 The winter mysteries

Hg(0) concentrations at TR remain at a fairly constant level of $0.98 \pm 0.06$ ng m$^{-3}$ in average
from April to August (Fig. 2b). This result is in good agreement with observations at
Neumayer (Ebinghaus et al., 2002). Pfaffhuber et al. (2012) attributed this phenomenon to the
lack of photochemical oxidation processes during the polar night. Conversely, Hg(0)
concentrations exhibit a gradual 30% decrease at DC, from $1.48 \pm 0.19$ in average in April to
$0.98 \pm 0.20$ ng m$^{-3}$ in August. This decreasing trend remains unexplained and possibly results
from the dry deposition of Hg(0) onto the snowpack (Angot et al., 2016b). In 2013,
measurements were performed at various height levels above the snow surface. Interestingly,
Angot et al. (2016b) reported a steeper decrease of Hg(0) concentrations close to the snow
surface suggesting that the snowpack may act as a sink for mercury. Similarly, a gradual 20%
decrease in Hg(0) concentrations is observed at DDU, from $0.94 \pm 0.07$ in average in April to
$0.72 \pm 0.10$ ng m$^{-3}$ in August (Fig. 2b). Based on an analysis of air mass back trajectories,
Angot et al. (2016a) suggested that this decreasing trend at DDU most likely results from
reactions occurring within the shallow boundary layer on the Antarctic plateau, subsequently
transported toward the coastal margins by katabatic winds. DDU is most of the time
influenced by inland air masses whereas several studies showed that stations such as NM are
not significantly impacted by air masses originating from the Antarctic plateau (Helmig et al.,
2007; Legrand et al., 2016b) explaining why concentrations remain rather stable at NM and
TR throughout winter.





Hg(0) concentration exhibits abrupt increases when moist and warm air masses from lower
latitudes occasionally reach the three ground-based Antarctic stations. At DDU, such events
are concomitant with an enhanced fraction of oceanic air masses reaching the site according
to the HYSPLIT model simulations, and with increased sodium concentrations (Angot et al.,
2016a). At DC, these advections of warm and moist air masses are confirmed by an increase
of temperature at 10 m a.g.l. and a high integrated water vapor column (Angot et al., 2016b).
Finally, based on a statistical analysis of source and sink regions, Pfaffhuber et al. (2012)
showed that transport from lower-latitude regions are frequently associated with the highest
Hg(0) concentrations at TR.
During the winter expedition ANTXXIX/6 on board R/V Polarstern over the Weddell Sea
(Fig. 1), Nerentorp Mastromonaco et al. (2016) observed depletions of Hg(0) characterized by
strong correlations with $O_3$. This is the first evidence of Hg(0) depletions occurring in winter.
The authors propose a dark mechanism involving $Br_2$. AMDEs in Antarctica are operationally
defined as Hg(0) concentrations below 0.60 ng m$^{-3}$ (Pfaffhuber et al., 2012). Based on this
threshold and on the $O_3$ signal, there is no evidence of Hg(0) depletions occurring during
months of complete darkness at the three ground-based Antarctic sites.
**3.2.1.2 Springtime AMDEs**
Before going further, it should be noted that TR is not a coastal station. It is located at an
elevation of 1275 m and approximately 220 km from the Antarctic coast. Contrarily, DDU is
located on a small island about one km offshore from the Antarctic mainland.
AMDEs are observed at TR in positive correlation with $O_3$ (r up to 0.56, $p$ value < 0.001,
Spearman test). Based on the 0.60 ng m$^{-3}$ threshold (see previous section), AMDEs occur in 2
% of the springtime observations, in line with the occurrence frequency of 5% reported by
Pfaffhuber et al. (2012) from February 2007 to June 2011. Based on a statistical analysis of
source and sink regions, Pfaffhuber et al. (2012) indicated that the spring Hg(0) sink, caused
by AMDEs, is mainly located within sea ice dense areas surrounding Queen Maud Land.
AMDEs at TR are weaker and less frequent when compared to the Arctic (see section 3.1.1.2)
likely partly due to the location of the station not being exposed directly to depletion events
but rather to transport of mercury-depleted air masses (Pfaffhuber et al., 2012). In contrast,
AMDEs occur in 28 % of the observations from 28 August to 5 October 2013 during the
spring expedition ANTXXIX/7 over sea ice areas of the Weddell Sea. At DDU, on the other
side of the Antarctic continent, data covering the spring period are scarce (Table 3). As
indicated by Angot et al. (2016a), the absence of depletions in spring 2012 tends to suggest



that AMDEs, if any, are not very frequent at DDU. Several studies reported a less efficient
bromine chemistry in East compared to West Antarctica due to a less sea-ice coverage (Theys
et al., 2011;  Legrand et al., 2016a). However, Angot et al. (2016a) reported low Hg(0)
concentrations ($0.71 \pm 0.11$ ng m$^{-3}$) and a significant positive correlation with O$_3$ (r up to 0.65,
*p* value < 0.0001, Spearman test) in springtime oceanic air masses, likely due to bromine
chemistry.

**3.2.1.3 Boundary layer dynamics on the Antarctic plateau in fall**

The fall maximum at DC likely partly results from a low boundary layer oxidative capacity
under low solar radiation limiting Hg(0) oxidation. Additionally, at DC, weak turbulence and
mixing, and strong temperature gradients near the surface are favored by light wind and clear
sky conditions (Argentini et al., 2013). The surface-based temperature inversions were
characterized by Pietroni et al. (2012) over the course of a year. In summer, a convective
boundary layer characterized by a maximum depth of 200-400 m (Argentini et al., 2005)
develops around midday. In winter, strong temperature inversions allow for a mixing depth of
a few tens of meters only. Based on the limited area model MAR (Modèle Atmosphérique
Régional), Angot et al. (2016b) indicated that the fall distinct maximum of Hg(0)
concentrations is concomitant with the time when the boundary layer lowers to ~ 50 m in
average and no longer exhibits a pronounced diurnal cycle. Hg(0) is thus suddenly dispersed
into a reduced volume of air, limiting the dilution. Similarly, several studies showed that NO$_x$
mixing ratios are enhanced when the boundary layer is shallow (Neff et al., 2008;  Frey et al.,

712   2013).

**3.2.1.4 Extremely active processes in summertime**

Summertime Hg(0) concentrations at the three ground-based sites exhibit a high variability
(Fig. 2b), suggesting extremely active processes at this time of the year. Undetected from
March to October, a diurnal cycle characterized by a noon Hg(0) maximum is observed in
summer at DDU and DC over the 2012-2015 period (Angot et al., 2016a;  Angot et al.,
2016b). At DC (DDU), Hg(0) concentrations range from ~ 0.6 ng m$^{-3}$  (~ 0.7 ng m$^{-3}$) on
average at night to ~ 1.0 ng m$^{-3}$ (~ 1.1 ng m$^{-3}$) on average around midday. Conversely, there is
no diurnal variation in Hg(0) concentrations at TR, in good agreement with observations
reported by Pfaffhuber et al. (2012) from February 2007 to June 2011. Similarly, there is no
mention of a daily cycle at NM, Terra Nova Bay, and McMurdo where summer campaigns
were carried out (Ebinghaus et al., 2002;  Temme et al., 2003;  Sprovieri et al., 2002;  Brooks
et al., 2008b). The absence of diurnal cycle at TR can be attributed to the absence of





sources/sinks for Hg(0) with a diurnal cycle in the vicinity of the site (Pfaffhuber et al., 2012).
The mean summertime Hg(0) concentration is significantly ($p$ value < 0.0001, Mann-Whitney
test) lower at DC (0.78 ± 0.46 ng m$^{-3}$) than at DDU (0.88 ± 0.32 ng m$^{-3}$) and TR (0.89 ± 0.29
ng m$^{-3}$), suggesting a more intense oxidation of Hg(0). The boundary layer oxidative capacity
has been shown to be high in summer on the Antarctic plateau with elevated levels of OH, O$_3$,
NO$_x$, and RO$_2$ radicals (Davis et al., 2001; Grannas et al., 2007; Eisele et al., 2008; Kukui et
al., 2014; Frey et al., 2015). Angot et al. (2016b) performed Hg(0) measurements in both the
atmospheric boundary layer and the interstitial air of the snowpack, and analyzed total
mercury in surface snow samples. The authors, in good agreement with Brooks et al. (2008a)
and Dommergue et al. (2012), suggested that the observed summertime Hg(0) diurnal cycle at
DC might be due to a dynamic daily cycle of Hg(0) oxidation, deposition to the snowpack,
and reemission from the snowpack. Similarly, a recent study (Wang et al., 2016) reported a
Hg(0) diurnal cycle at Kunlun station (80°25'S, 77°6'E) located near Dome A (80°22'S,
77°27'E) – the highest elevation point on the Antarctic plateau (4090 m). This suggests that
the dynamic daily cycle of Hg(0) oxidation, deposition to the snowpack, and reemission from
the snowpack probably occurs throughout the Antarctic plateau. Based on an analysis of air
mass back trajectories, Angot et al. (2016a) showed that measurements at DDU on the East
Antarctic coast are dramatically influenced by air masses exported from the Antarctic Plateau
by strong katabatic winds. The advection of inland air masses enriched in oxidants – NO$_x$, O$_3$,
and OH (Grilli et al., 2013; Kukui et al., 2012) – and Hg(II) species likely results in the build-
up of an atmospheric reservoir of Hg(II) species at DDU, as supported by elevated levels of
total mercury in surface snow samples (Angot et al., 2016a). The diurnal cycle observed at
DDU – regardless of wind speed and direction – might result from a local dynamic cycle of
oxidation/deposition/reemission in the presence of elevated levels of Hg(II) species along
with emissions of mercury from ornithogenic soils – formed by an accumulation of penguin
excreta.
Hg(0) depletion events occur each year in summer at DC with Hg(0) concentrations
remaining low (~ 0.40 ng m$^{-3}$) for several weeks. These depletion events do not resemble to
the ones observed in the Arctic. They are not associated with depletions of O$_3$, and occur as
air masses stagnate over the Plateau which could favor an accumulation of oxidants within the
shallow boundary layer (Angot et al., 2016b). At TR, Pfaffhuber et al. (2012) reported
episodic low Hg(0) concentrations in summer, anti-correlated with O$_3$, and associated with the
transport of inland air masses. Results at TR (Pfaffhuber et al., 2012) and DDU (Angot et al.,



2016a), along with observations from earlier studies at other coastal Antarctic sites (Sprovieri
et al., 2002; Temme et al., 2003), demonstrate that the inland atmospheric reservoir can
influence the cycle of atmospheric mercury at a continental scale, especially in areas
influenced by recurrent katabatic winds.
Additionally, Pfaffhuber et al. (2012) indicated that the ocean is a source of mercury to TR.
Similarly, at DDU, Angot et al. (2016a) reported elevated (1.04 ± 0.29 ng m$^{-3}$) Hg(0)
concentrations in oceanic air masses along with a significant positive correlation between
Hg(0) and the daily-averaged percentage of oceanic air masses (r = 0.50, $p$ value < 0.0001,
Spearman test). These results are in line with the summer Hg(0) enhancement in the Arctic
likely partly due to oceanic evasion from ice-free open waters (see section 3.1.1.3).
**3.2.2  Comparison with models**
Table 4 displays goodness-of-fit statistics between monthly-averaged modeled and observed
data in 2013. ECHMERIT slightly underestimates Hg(0) concentrations at the three ground-
based Antarctic sites. Contrarily, the three other global models overestimate Hg(0) levels,
suggesting an underestimation of sinks. The ability of the four models to reproduce the
observed seasonality of Hg(0) concentrations at ground-based Antarctic sites in 2013 is
shown in Fig. 6b and discussed in the following sections. The inter-annual variability of the
monthly Hg(0) concentration distribution at Antarctic ground-based sites as simulated by
GEM-MACH-Hg and GEOS-Chem is displayed in Fig. 7b while Table 5 shows the percent
bias between hourly-averaged modeled and observed data on a seasonal basis from 2011 to

778    2014.

**3.2.2.1 Seasonal variation**
**a) Winter**
GEOS-Chem, GEM-MACH-Hg, and GLEMOS overestimate year 2013 Hg(0) concentrations
in winter at the three ground-based stations (Fig. 6a). This trend repeats year after year for
GEOS-Chem and GEM-MACH-Hg (Table 5). The most striking result, however, is the
modeled gradual increase of Hg(0) concentrations over the course of winter at the three
ground-based sites according to ECHMERIT, GEOS-Chem, and GEM-MACH-Hg. A mean
gradual increase of 9 %, 19 %, and 11 % is predicted by the three models, respectively, from
May to August. GLEMOS, however, predicts a mean gradual decrease of 5 % over the course
of winter at the three sites. It is to be noted (see section 3.2.1.1) that Hg(0) concentrations are
constant from May to August at TR, exhibit a gradual 30 % decrease at DC possibly due to



the dry deposition of Hg(0), and a gradual 20 % decrease at DDU due to advection of inland
air masses. All in all, the four models misrepresent the decreasing trend at DC and DDU. This
might be due to several factors including underestimation of concentrations of oxidants over
the East Antarctic plateau at this period of the year, omission of heterogeneous mechanisms,
and significant bias in Southern Hemisphere emissions, including oceanic evasion. The strong
increase (19 %) of Hg(0) concentrations from May to August predicted by GEOS-Chem is not
restricted to the Antarctic continent but is obtained for the whole Southern Hemisphere (Fig. 3
in Song et al., 2015). The emission inversion performed by Song et al. (2015) overturns the
seasonality of oceanic emissions and better reproduces the ground-based Hg(0) observations
in the Southern Hemisphere mid-latitudes and at TR. Further work, including sensitivity tests,
is needed to explain the discrepancies between observed and modeled trends.
Additionally, all of the four models are unable to capture the differences in trends observed at
the three ground-based sites (constant *vs.* decreasing concentrations). As discussed in section
3.2.1.1, TR, contrarily to DDU, is not significantly influenced by inland air masses. This
large-scale airflow pattern will have to be captured by models in order to better reproduce
observations. Interestingly, Zatko et al. (2016) calculated the annual mean surface wind
convergence/divergence over the Antarctic continent using GEOS-Chem. The results –
consistent with those by Parish and Bromwich (1987) and Parish and Bromwich (2007) –
correctly indicate that the large-scale airflow pattern in Antarctica flows from the East
Antarctic plateau towards the coastal margins and accurately highlight major regions of wind
convergence. The findings from this study can be used as the basis for future research.
**b) Spring**
Based on the 0.60 ng m$^{-3}$ threshold, GEM-MACH-Hg and GEOS-Chem do not predict any
AMDE at TR over the 2011-2014 period. Considering the low occurrence frequency based on
observations (2 %, see section 3.2.1.2), this result is not unreasonable. Similarly, GEM-
MACH-Hg does not predict any AMDE at DDU. However, GEOS-Chem predicts AMDEs in
1.5 % of the springtime observations at DDU. This over-prediction of AMDEs at DDU likely
results from the constant value of 5 pptv of BrO added in the springtime Antarctic boundary
layer. While Saiz-Lopez et al. (2007) reported a spring maximum of up to 7 pptv at Halley
Station (75°35'S, 26°30'W, West Antarctic coast), Legrand et al. (2016a) suggested a BrO
mixing ratio ≤ 1 pptv at DDU (East Antarctic coast) in spring using an off-line chemistry
transport model. Based on the oxygen and nitrogen isotope analysis of airborne nitrate,
Savarino et al. (2007) provided further evidence for low BrO levels in the vicinity of DDU.





**c) Fall**
None of the four models capture the fall maximum at DC (Fig. 6b). While a spatially and
temporally resolved distribution of concentrations of oxidants on the East Antarctic Plateau is
needed, the boundary layer dynamics must also be taken into account. Based on the work by
Lin and McElroy (2010), Zatko et al. (2016) incorporated a calculation of the boundary layer
height across Antarctica and Greenland into GEOS-Chem. One could also rely on model
outputs from the limited area model MAR, validated against observations at DC (Gallée and
Gorodetskaya, 2010; Gallée et al., 2015). This model agrees very well with observations and
provides reliable and useful information about surface turbulent fluxes, vertical profiles of
vertical diffusion coefficients and boundary layer height.
**d) Summer**
The daily variation of Hg(0) concentrations was investigated based on hourly-averaged data
provided by GEOS-Chem and GEM-MACH-Hg. The two models are not able to reproduce
the noon maximum observed at DC and DDU in summer (3.2.1.4), suggesting that the
dynamic daily cycle of deposition and reemission at the air/snow interface is not captured by
the models. The bidirectional exchange of Hg(0) is complex and influenced by multiple
environmental variables (e.g., UV intensity, temperature, atmospheric turbulence, presence of
reactants) limiting the accuracy of flux modeling (Zhu et al., 2016). The work carried out by
Durnford et al. (2012) in the Arctic and by Zatko et al. (2016) in Antarctica could be good
starting points for future research. The former developed a new dynamic physically-based
snowpack model to determine the fate of mercury deposited onto snowpacks; the latter
incorporated an idealized snowpack along with a snow radiative transfer model (Zatko et al.,
2013) into GEOS-Chem to investigate the impact of snow nitrate photolysis on the boundary
layer chemistry across Antarctica.
**3.2.2.2 Reactive mercury and deposition**
According to Fig. 8b, ECHMERIT predicts low RM concentrations during the whole 2013
year at the three ground-based stations (annual averages of 10, 7, and 6 pg m$^{-3}$ at TR, DC, and
DDU, respectively). GEOS-Chem predicts a peak in spring at the three sites (up to ~ 160 pg
m$^{-3}$ in average October at DC), and quite elevated concentrations in summer and fall (~ 85 pg
m$^{-3}$ in average). GEM-MACH-Hg predicts increased concentrations in summer at TR and
DDU only. Finally, GLEMOS predicts a more intense summer peak at DC (up to ~ 130 pg m$^{-}$
$^{3}$ in average in November) than at DDU and TR. Measurements of RM are scarce in
Antarctica and have never been reported on a year-round basis. RM concentrations ranging
from 100 to 1000 pg m$^{-3}$ have been reported in summer at South Pole (Brooks et al., 2008a)
and several studies have reported elevated concentrations at coastal sites in spring during the
AMDEs season (165 pg m$^{-3}$ in average at Mc Murdo (Brooks et al., 2008b)) and in summer
(mean RGM concentration of 116 pg m$^{-3}$ at Terra Nova Bay (Sprovieri et al., 2002); RGM
and Hg(p) concentrations ranging from 5 to > 300 pg m$^{-3}$ and from 15 to 120 pg m$^{-3}$,
respectively, at Neumayer (Temme et al., 2003)). These results along with the seasonal
pattern of Hg(0) reported in section 3.2.1 suggest that the atmospheric boundary layer is
enriched in RM in summer, especially on the Antarctic plateau, and that the four models tend
to underestimate the summertime concentrations. Year-round measurements are needed to
further evaluate the accuracy of models predictions.
The total (wet + dry) deposition flux for year 2013 is equal to 1.0, 3.3, 2.5, and 3.9 µg m$^{-2}$ yr$^{-1}$
at TR, 0.8, 1.5, 0.8, and 1.1 µg m$^{-2}$ yr$^{-1}$ at DC, and 4.3, 9.7, 9.7, and 4.1 µg m$^{-2}$ yr$^{-1}$ at DDU
according to GLEMOS, GEOS-Chem, GEM-MACH-Hg, and ECHMERIT, respectively.
Deposition during summertime accounts for 73, 53, 68, and 35 % of the total deposition at
TR, 58, 50, 37, and 35 % at DC, and 58, 61, 89, and 28 % at DDU according to GLEMOS,
GEOS-Chem, GEM-MACH-Hg, and ECHMERIT, respectively. There are no measurements
of wet and dry deposition in Antarctica, except Angot et al. (2016b) who reported a Hg(0) dry
deposition velocity of 9.3 10$^{-5}$ cm s$^{-1}$ in winter at DC. Similarly to the Arctic (see section
3.1.2.2), deposition of mercury is typically inferred from concentrations of total mercury in
the snowpack. To the best of our knowledge, results found in Angot et al. (2016b) are the only
reported over various seasons. Higher total mercury concentrations in surface snow samples
in summer suggest an enhanced deposition at this period of the year. Alternatively, deposition
of mercury can be inferred from the biomonitoring of Antarctic macrolichens and mosses.
Large-scale and long-term biomonitoring surveys of mercury deposition have been performed
in Victoria Land (Bargagli et al., 1993; Bargagli et al., 2005). While all four models predict
higher total mercury deposition for year 2013 at high Arctic (ALT, SND, NYA) *vs.* Antarctic
ground-based sites, significantly higher mercury concentrations in Antarctic *vs.* Northern
Hemisphere lichens suggest otherwise (Bargagli et al., 1993).
Wet deposition accounts for 14, 53, 47, and 0 % of the total (wet + dry) flux at TR, 35, 7, 14,
and 0 % at DC, and 68, 57, 60, and 8 % at DDU according to GLEMOS, GEOS-Chem, GEM-
MACH-Hg, and ECHMERIT, respectively. The amount of precipitation is equal to 214, 242,
291, and 1127 mm yr$^{-1}$ at TR, 33, 29, 24, and 60 mm yr$^{-1}$ at DC, and 643, 792, 895, and 1751





mm yr$^{-1}$ at DDU according to GLEMOS, GEOS-Chem, GEM-MACH-Hg, and ECHMERIT,
respectively. Ground-based measurements of precipitation are sparse and difficult to obtain in
Antarctica. Strong winds in coastal regions make it difficult to tell the difference between
blowing snow and precipitation (Palerme et al., 2014). On the Antarctic plateau, a significant
part of the precipitation falls in the form of ice crystals (diamond dust) under clear-sky
conditions (Bromwich, 1988; Fujita and Abe, 2006). Satellite observations of precipitation in
Antarctica by active sensors are now possible (Liu, 2008; Stephens et al., 2008). According
to Palerme et al. (2014), the mean annual snowfall rate is < 20 mm water equivalent yr$^{-1}$ at
DC, ranges from 20 to 100 mm yr$^{-1}$ at TR, and from 500 to 700 mm yr$^{-1}$ at DDU. The low
amount of precipitation at DC might, however, be offset by the high mercury-capture
efficiency of ice crystals (Douglas et al., 2008) that are frequently observed at that site
(Bromwich, 1988; Fujita and Abe, 2006).

## 4  Summary and future perspectives

The data compiled in this study represent the latest available in Polar Regions. While the
Arctic is a semi-enclosed ocean almost completely surrounded by land, Antarctica is a land
mass – covered with an immense ice shelf – surrounded by ocean. Therefore, the cycle of
atmospheric mercury in the two regions presents both similarities and differences. Springtime
AMDEs are observed in both regions at coastal sites (see sections 3.1.1.2 and 3.2.1.2). Their
frequency and magnitude depend on parameters such as sea-ice dynamics, temperature, and
concentration of bromine species, and exhibit a significant but poorly understood inter-annual
variability. Additionally, coastal sites in the two regions are influenced by both snowpack
reemission and oceanic evasion of Hg(0) in summer (see sections 3.1.1.3 and 3.2.1.4). As
evidenced in section 3.1.1.3, the summertime enhancement of Hg(0) concentrations exhibits a
significant but little understood inter-annual variability at Arctic sites. The cycle of
atmospheric mercury differs between the Arctic and Antarctica, primarily because of their
different geography. Arctic sites are significantly influenced by mercury emissions from
Northern Hemisphere mid-latitudes – especially in winter (see section 3.1.1.1). Coastal
Antarctic sites are significantly influenced by the reactivity of atmospheric mercury observed
on the Antarctic Plateau due to the large-scale airflow pattern flowing from the East Antarctic
ice sheet towards the coastal margins (katabatic winds). As discussed in section 3.2, the cycle
of atmospheric mercury on the Antarctic Plateau is surprising and involves yet unraveled





mechanisms in winter and a daily bidirectional exchange of Hg(0) at the air/snow interface in
summer.

From the comparison of multi-model simulations with observations, we identified whether the
processes that affect Hg(0) seasonality and inter-annual variability, including mercury
oxidation, deposition and reemission, are appropriately understood and represented in the
models. Generally, models reproduce quite fairly the observed seasonality at Arctic sites but
fail to reproduce it at Antarctic sites. In order for the models to reproduce the seasonality of
Hg(0) concentrations in Antarctica, parameterization of the boundary layer dynamics (see
section 3.1.1.3) and of the large-scale airflow pattern (see above) is needed. Moreover,
reaction pathways might be missing or inappropriately incorporated in models. Heterogeneous
reactions, although poorly understood (Subir et al., 2012), might be required to explain the
reactivity on the Antarctic Plateau. Additionally, while $NO_x$ chemistry was shown to prevail
upon halogens chemistry in East Antarctica in summer (Legrand et al., 2009; Grilli et al.,
2013) it is currently incorporated in none of the four global models.

Based on this study, the following research gaps need to be addressed:

1. Improving the spatial resolution of RM measurements. There is presently no year-round
data available in Antarctica. The Tekran speciation unit suffers from significant biases and
interferences, is expensive, labor-intensive, and requires trained operators. Passive samplers,
such as Polyethersulfone cation exchange membranes, could provide an alternative (Huang et
al., 2014).

2. Unraveling of Hg(II) speciation. The exact speciation – expected to vary with space and
time – remains unknown. Identification of Hg(II) species in ambient air emerges as one of the
priorities for future research (Gustin et al., 2015). Recent advancement on analytical
techniques may offer new insights into Hg(II) speciation (Huang et al., 2013; Jones et al.,
2016). However, further research is still needed and application of passive samplers for
collection and identification of Hg(II) compounds should be tested in various environments
and at different times of the year. Such advancement will greatly improve our understanding
of atmospheric redox processes.

3. Improving the spatial resolution of measurements of total mercury in snow samples. These
measurements are an alternative to wet and dry deposition measurements – difficult to
perform in Polar Regions.



4. Investigation of the fundamental environmental processes driving the inter-annual
variability of Hg(0) concentrations, especially at Arctic sites. Further work is needed to
establish the degree to which temperature and sea-ice dynamics across the Arctic alters
mercury chemistry in spring and summer. This will also open up new opportunities to explore
the influence of Climate Change on the cycle of mercury in Polar Regions.
5. Investigation (and quantification) of the oceanic fluxes of Hg(0) during oceanographic
campaigns across the Arctic and Austral Oceans. This will largely reduce the uncertainty in
the mercury budget estimation in Polar Regions.
6. Reducing uncertainties in existing kinetic parameters and quantitatively investigate the
effect of temperature on the rate constants (Subir et al., 2011). Limited data are available for
temperature applicable to atmospheric conditions, especially in Polar Regions. Achieving this
will largely reduce uncertainties in atmospheric models.
7. Investigation of the influence of atmospheric surfaces (e.g., aerosols, clouds, ice, snow
covers, ice crystals). This is a major gap for adequate modeling of mercury cycling (Subir et
al., 2012) and studies addressing this are critically needed.

**Acknowledgements**
HA, OM, and AD thank the overwintering crew: S. Aguado, D. Buiron, N. Coillard, G.
Dufresnes, J. Guilhermet, B. Jourdain, B. Laulier, S. Oros, A. Thollot, and N. Vogel at DDU,
S. Aubin, A. Barbero, N. Hueber, C. Lenormant, and R. Jacob at DC. This work contributed
to the EU-FP7 project Global Mercury Observation System (GMOS, www.gmos.eu) and has
been supported by a grant from Labex OSUG@2020 (Investissements d'avenir – ANR10
LABX56), and the Institut Universitaire de France. Logistical and financial support was
provided by the French Polar Institute IPEV (Program 1028, GMOstral). KAP thanks the
Norwegian Environmental Agency and the Norwegian Antarctic Research Expeditions for
long-term financial support of Norwegian mercury measurements and in particular the
technicians J.H. Wasseng and A. Bäcklund at NILU for their excellent care taking of the
Tekran monitors. NES and SS acknowledge support from the U.S. National Science
Foundation Atmospheric Chemistry Program under grant #1053648.





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





| | Code | Elevation (m a.s.l.) | Analyte | Instrumentation | Flow rate (L/min) | Resolution | Filter at the inlet | Sampling line |
|---|---|---|---|---|---|---|---|---|
| **Arctic** | ALT | 195 | Hg(0) | Tekran 2537A | 1.0 | 5 min | speciation unit | heated |
| | SND | 30 | Hg(p), RGM | Tekran 2537A/1130/1135 | 10.0 | 2 h | | heated |
| | NYA | 474 | Hg(0) | Tekran 2537A | 1.5 | 5 min | sodalime | heated |
| | AND | 10 | Hg(0) | Tekran 2537A | 1.5 | 5 min | 2 μm PTFE and sodalime | heated |
| | | | Hg(0) | Tekran 2537A | 1.5 | 5 min | 2 μm PTFE and sodalime | heated |
| **Antarctica** | TR | 1275 | Hg(0) | Tekran 2537A | 1.5 | 5 min | 2 μm PTFE filter | unheated |
| | DC | 3220 | Hg(0) | Tekran 2537A | 0.8 | 5-15 min | 0.45 μm PTFE filter | unheated |
| | DDU | 43 | Hg(0) | Tekran 2537B | 1.0 | 10-15 min | 0.20 μm PTFE filter | unheated |
| | ANT | 20 | Hg(0) | Tekran 2537A | 1.0 | 5 min | speciation unit | heated |
| | | | Hg(p), RGM | Tekran 2537A/1130/1135 | 10.0 | 2 h | | heated |
| | OSO | 15 | Hg(0) | Tekran 2537A | 1.0 | 5 min | 0.45 μm PTFE filter | unheated |

**Table 1:** Summary of the instrumentation used at the various Polar sites to measure atmospheric mercury species.



|  | Station | 2011 | | | | 2012 | | | | 2013 | | | | 2014 | | | | 2015 | | | |
|---|---|---|---|---|---|---|---|---|---|---|---|---|---|---|---|---|---|---|---|---|---|
|  |  | n | mean | median | SD | n | mean | median | SD | n | mean | median | SD | n | mean | median | SD | n | mean | median | SD |
| Arctic | ALT | 8040 | 1.39 | 1.35 | 0.45 | 8447 | 1.21 | 1.21 | 0.35 | 8048 | 1.31 | 1.39 | 0.46 | 8358 | 1.45 | 1.45 | 0.33 | na | na | na | na |
| | SND | 4712 | 1.26 | 1.34 | 0.32 | 7932 | 1.44 | 1.44 | 0.26 | 6605 | 1.57 | 1.49 | 0.44 | 4991 | 1.36 | 1.36 | 0.35 | 1059 | 1.11 | 1.11 | 0.32 |
| | NYA | 8173 | 1.51 | 1.59 | 0.31 | 8181 | 1.51 | 1.54 | 0.21 | 6980 | 1.47 | 1.52 | 0.30 | 6730 | 1.48 | 1.57 | 0.33 | 8342 | 1.49 | 1.49 | 0.21 |
| | AND | 7444 | 1.61 | 1.61 | 0.15 | 8428 | 1.61 | 1.61 | 0.13 | 7862 | 1.53 | 1.56 | 0.15 | 8146 | 1.50 | 1.51 | 0.16 | 7146 | 1.50 | 1.50 | 0.10 |
| Antarctica | TR | 5978 | 0.95 | 0.99 | 0.20 | 7808 | 0.98 | 0.97 | 0.15 | 8197 | 0.90 | 0.93 | 0.15 | 7421 | 0.95 | 1.00 | 0.21 | 3670 | 0.94 | 0.93 | 0.31 |
| | DC | na | na | na | na | 3761 | 0.76 | 0.70 | 0.24 | 2900 | 0.84 | 0.87 | 0.27 | na | na | na | na | 8383 | 1.06 | 1.12 | 0.41 |
| | DDU | na | na | na | na | 5949 | 0.91 | 0.92 | 0.20 | 5121 | 0.85 | 0.85 | 0.19 | 1958 | 0.85 | 0.82 | 0.38 | 3114 | 0.86 | 0.87 | 0.19 |

**Table 2:** Annually-based statistics (number of hourly-averaged data (n), mean, median, standard deviation (SD)) of Hg(0) concentrations (in ng m$^{-3}$) at ground-based Polar sites over the 2011-2015 period. Note that 2013 data at DC refer to concentrations recorded at 210 cm above the snowpack. The 2015 data coverage is May to June at SND and January to May at DDU (see Table 3). na: not available due to QA/QC invalidation, instrument failure, or because the QA/QC validation is currently in progress (2015 data).



| | ALT | | | | SND | | | | NYA | | | | AND | | | | TR | | | | DC | | | | DDU | | | |
|---|---|---|---|---|---|---|---|---|---|---|---|---|---|---|---|---|---|---|---|---|---|---|---|---|---|---|---|---|
| | n | mean | median | SD | n | mean | median | SD | n | mean | median | SD | n | mean | median | SD | n | mean | median | SD | n | mean | median | SD | n | mean | median | SD |
| **2011** | | | | | | | | | | | | | | | | | | | | | | | | | | | | |
| Jan | 736 | 1.44 | 1.44 | 0.06 | 698 | 1.46 | 1.46 | 0.07 | 739 | 1.49 | 1.58 | 0.26 | 627 | 1.72 | 1.71 | 0.14 | 671 | 0.85 | 0.86 | 0.25 | na | na | na | na | na | na | na | na |
| Feb | 664 | 1.35 | 1.39 | 0.22 | 631 | 1.40 | 1.40 | 0.08 | 661 | 1.48 | 1.55 | 0.31 | 446 | 1.72 | 1.73 | 0.14 | 656 | 0.98 | 1.06 | 0.25 | na | na | na | na | na | na | na | na |
| Mar | 740 | 1.33 | 1.35 | 0.33 | 613 | 1.24 | 1.30 | 0.28 | 548 | 1.43 | 1.59 | 0.38 | 673 | 1.71 | 1.73 | 0.21 | 735 | 1.06 | 1.05 | 0.11 | na | na | na | na | na | na | na | na |
| Apr | 720 | 0.87 | 0.91 | 0.52 | 621 | 1.05 | 1.13 | 0.36 | 719 | 1.58 | 1.65 | 0.31 | 631 | 1.59 | 1.59 | 0.16 | 711 | 1.01 | 1.01 | 0.05 | na | na | na | na | na | na | na | na |
| May | 647 | 1.38 | 1.28 | 0.73 | 622 | 0.91 | 0.91 | 0.44 | 709 | 1.17 | 1.18 | 0.42 | 494 | 1.42 | 1.41 | 0.13 | 718 | 0.99 | 0.99 | 0.03 | na | na | na | na | na | na | na | na |
| Jun | 690 | 1.87 | 1.85 | 0.31 | 434 | 1.27 | 1.21 | 0.35 | 716 | 1.46 | 1.58 | 0.28 | 658 | 1.53 | 1.53 | 0.11 | 614 | 0.98 | 0.99 | 0.04 | na | na | na | na | na | na | na | na |
| Jul | 672 | 1.96 | 1.97 | 0.48 | na | na | na | na | 647 | 1.77 | 1.73 | 0.27 | 676 | 1.55 | 1.54 | 0.09 | 733 | 0.98 | 0.98 | 0.05 | na | na | na | na | na | na | na | na |
| Aug | 724 | 1.62 | 1.63 | 0.19 | na | na | na | na | 663 | 1.66 | 1.68 | 1.15 | 606 | 1.52 | 1.53 | 0.08 | 169 | 0.92 | 0.92 | 0.04 | na | na | na | na | na | na | na | na |
| Sep | 670 | 1.21 | 1.20 | 0.06 | 458 | 1.30 | 1.30 | 0.04 | 715 | 1.66 | 1.66 | 0.12 | 444 | 1.58 | 1.59 | 0.09 | na | na | na | na | na | na | na | na | na | na | na | na |
| Oct | 719 | 1.16 | 1.16 | 0.02 | 107 | 1.23 | 1.23 | 0.03 | 669 | 1.59 | 1.60 | 0.12 | 728 | 1.62 | 1.64 | 0.10 | na | na | na | na | na | na | na | na | na | na | na | na |
| Nov | 395 | 1.20 | 1.21 | 0.06 | na | na | na | na | 681 | 1.32 | 1.34 | 0.30 | 719 | 1.64 | 1.62 | 0.11 | 254 | 0.59 | 0.71 | 0.34 | na | na | na | na | na | na | na | na |
| Dec | 663 | 1.29 | 1.30 | 0.06 | 528 | 1.52 | 1.53 | 0.05 | 706 | 1.59 | 1.59 | 0.06 | 742 | 1.71 | 1.71 | 0.06 | 717 | 0.87 | 0.86 | 0.29 | na | na | na | na | na | na | na | na |
| **2012** | | | | | | | | | | | | | | | | | | | | | | | | | | | | |
| Jan | 595 | 1.33 | 1.36 | 0.10 | 744 | 1.53 | 1.53 | 0.07 | 595 | 1.62 | 1.61 | 0.06 | 720 | 1.75 | 1.74 | 0.07 | 497 | 1.07 | 1.08 | 0.28 | 259 | 0.61 | 0.57 | 0.33 | 576 | 1.06 | 1.09 | 0.32 |
| Feb | 685 | 1.32 | 1.33 | 0.07 | 696 | 1.48 | 1.49 | 0.07 | 696 | 1.59 | 1.59 | 0.06 | 696 | 1.76 | 1.75 | 0.05 | 660 | 1.03 | 1.00 | 0.23 | 593 | 0.93 | 1.00 | 0.42 | 670 | 1.01 | 1.03 | 0.23 |
| Mar | 722 | 0.92 | 1.02 | 0.41 | 744 | 1.26 | 1.35 | 0.29 | 726 | 1.48 | 1.59 | 0.28 | 744 | 1.73 | 1.73 | 0.08 | 744 | 0.97 | 0.97 | 0.05 | 67 | 1.14 | 1.14 | 0.26 | 635 | 0.97 | 0.95 | 0.09 |
| Apr | 695 | 0.79 | 0.75 | 0.49 | 319 | 1.29 | 1.32 | 0.41 | 550 | 1.31 | 1.45 | 0.37 | 720 | 1.59 | 1.60 | 0.12 | 712 | 0.97 | 0.97 | 0.04 | na | na | na | na | 696 | 0.92 | 0.94 | 0.11 |
| May | 698 | 1.19 | 1.27 | 0.59 | 703 | 1.58 | 1.63 | 0.52 | 697 | 1.39 | 1.46 | 0.26 | 744 | 1.55 | 1.59 | 0.16 | 649 | 0.97 | 0.97 | 0.03 | na | na | na | na | 663 | 0.88 | 0.88 | 0.08 |
| Jun | 720 | 1.52 | 1.52 | 0.24 | 719 | 1.61 | 1.60 | 0.22 | 698 | 1.52 | 1.50 | 0.10 | 720 | 1.56 | 1.57 | 0.09 | 654 | 0.95 | 0.94 | 0.10 | 423 | 0.82 | 0.81 | 0.06 | 663 | 0.88 | 0.88 | 0.08 |
| Jul | 728 | 1.50 | 1.44 | 0.33 | 744 | 1.61 | 1.59 | 0.22 | 734 | 1.68 | 1.68 | 0.17 | 412 | 1.61 | 1.61 | 0.07 | 487 | 0.87 | 0.87 | 0.06 | 624 | 0.70 | 0.70 | 0.05 | 101 | 0.79 | 0.79 | 0.07 |
| Aug | 744 | 1.27 | 1.26 | 0.09 | 593 | 1.54 | 1.53 | 0.12 | 678 | 1.70 | 1.69 | 0.09 | 744 | 1.52 | 1.52 | 0.06 | 670 | 1.01 | 1.02 | 0.07 | 682 | 0.66 | 0.67 | 0.05 | 107 | 0.63 | 0.62 | 0.05 |
| Sep | 657 | 1.16 | 1.16 | 0.06 | 631 | 1.43 | 1.42 | 0.07 | 713 | 1.58 | 1.56 | 0.10 | 720 | 1.46 | 1.45 | 0.07 | 612 | 1.08 | 1.08 | 0.08 | 591 | 0.72 | 0.66 | 0.14 | 131 | 0.99 | 1.00 | 0.09 |
| Oct | 742 | 1.16 | 1.16 | 0.04 | 601 | 1.28 | 1.27 | 0.06 | 664 | 1.38 | 1.39 | 0.05 | 744 | 1.56 | 1.56 | 0.10 | 744 | 1.02 | 1.01 | 0.12 | 431 | 0.79 | 0.81 | 0.20 | 719 | 0.82 | 0.84 | 0.14 |
| Nov | 718 | 1.16 | 1.17 | 0.06 | 694 | 1.31 | 1.28 | 0.09 | 700 | 1.40 | 1.41 | 0.08 | 720 | 1.57 | 1.57 | 0.07 | 699 | 0.94 | 0.90 | 0.15 | na | na | na | na | 428 | 0.76 | 0.74 | 0.24 |
| Dec | 743 | 1.16 | 1.18 | 0.05 | 744 | 1.29 | 1.27 | 0.11 | 730 | 1.45 | 1.47 | 0.15 | 744 | 1.70 | 1.67 | 0.09 | 680 | 0.90 | 0.88 | 0.22 | na | na | na | na | 555 | 0.82 | 0.8 | 0.21 |
| **2013** | | | | | | | | | | | | | | | | | | | | | | | | | | | | |
| Jan | 468 | 1.25 | 1.27 | 0.12 | 729 | 1.5 | 1.51 | 0.13 | 483 | 1.52 | 1.54 | 0.13 | 717 | 1.66 | 1.66 | 0.05 | 711 | 0.97 | 0.96 | 0.24 | 762 | 0.69 | 0.64 | 0.30 | 644 | 0.88 | 0.84 | 0.37 |
| Feb | 671 | 1.23 | 1.27 | 0.14 | 378 | 1.46 | 1.45 | 0.06 | 596 | 1.65 | 1.67 | 0.10 | 671 | 1.68 | 1.67 | 0.06 | 665 | 0.93 | 0.97 | 0.21 | 585 | 0.68 | 0.59 | 0.41 | 450 | 0.81 | 0.81 | 0.23 |
| Mar | 664 | 1.14 | 1.28 | 0.40 | na | na | na | na | 671 | 1.39 | 1.45 | 0.30 | 725 | 1.57 | 1.59 | 0.07 | 727 | 0.98 | 1.00 | 0.08 | 487 | 1.16 | 1.15 | 0.19 | 215 | 0.81 | 0.77 | 0.15 |
| Apr | 707 | 0.65 | 0.60 | 0.49 | 582 | 1.43 | 1.38 | 0.63 | 689 | 1.22 | 1.40 | 0.51 | 680 | 1.46 | 1.49 | 0.20 | 704 | 0.98 | 0.97 | 0.05 | 271 | 1.16 | 1.14 | 0.16 | 635 | 0.96 | 0.95 | 0.04 |
| May | 739 | 0.91 | 0.87 | 0.67 | 744 | 1.39 | 1.23 | 0.75 | 744 | 1.40 | 1.48 | 0.33 | 732 | 1.41 | 1.40 | 0.10 | 688 | 0.94 | 0.94 | 0.03 | 464 | 1.01 | 0.99 | 0.10 | 725 | 0.88 | 0.88 | 0.04 |
| Jun | 696 | 1.43 | 1.53 | 0.59 | 719 | 1.89 | 1.96 | 0.50 | 686 | 1.45 | 1.63 | 0.42 | 713 | 1.41 | 1.43 | 0.13 | 718 | 0.95 | 0.95 | 0.02 | 297 | 0.93 | 0.93 | 0.05 | 661 | 0.83 | 0.83 | 0.05 |
| Jul | 742 | 1.82 | 1.80 | 0.23 | 709 | 1.97 | 1.95 | 0.28 | 206 | 1.52 | 1.50 | 0.13 | 717 | 1.43 | 1.45 | 0.12 | 713 | 0.96 | 0.96 | 0.03 | 554 | 0.89 | 0.89 | 0.05 | 639 | 0.80 | 0.81 | 0.09 |
| Aug | 720 | 1.71 | 1.67 | 0.15 | 538 | 1.84 | 1.76 | 0.23 | 716 | 1.63 | 1.60 | 0.12 | 622 | 1.42 | 1.43 | 0.11 | 679 | 0.90 | 0.91 | 0.05 | 591 | 0.75 | 0.75 | 0.08 | 655 | 0.73 | 0.73 | 0.10 |
| Sep | 720 | 1.43 | 1.41 | 0.04 | 412 | 1.47 | 1.49 | 0.13 | 690 | 1.51 | 1.51 | 0.07 | 266 | 1.50 | 1.49 | 0.08 | 670 | 0.87 | 0.87 | 0.06 | 616 | 0.85 | 0.85 | 0.08 | 82 | 0.68 | 0.68 | 0.06 |
| Oct | 744 | 1.36 | 1.36 | 0.04 | 502 | 1.41 | 1.40 | 0.05 | 687 | 1.48 | 1.48 | 0.06 | 739 | 1.56 | 1.56 | 0.05 | 710 | 0.79 | 0.76 | 0.12 | 245 | 0.75 | 0.79 | 0.17 | na | na | na | na |
| Nov | 605 | 1.36 | 1.36 | 0.02 | 597 | 1.40 | 1.39 | 0.07 | 298 | 1.40 | 1.41 | 0.07 | 545 | 1.60 | 1.61 | 0.05 | 606 | 0.76 | 0.76 | 0.18 | 431 | 0.66 | 0.60 | 0.33 | na | na | na | na |
| Dec | 646 | 1.32 | 1.33 | 0.07 | 694 | 1.36 | 1.36 | 0.09 | 514 | 1.52 | 1.44 | 0.31 | 735 | 1.69 | 1.69 | 0.05 | 606 | 0.78 | 0.76 | 0.20 | 213 | 0.84 | 0.85 | 0.24 | 415 | 0.98 | 0.97 | 0.25 |
| **2014** | | | | | | | | | | | | | | | | | | | | | | | | | | | | |
| Jan | 743 | 1.47 | 1.47 | 0.07 | 719 | 1.41 | 1.37 | 0.16 | 701 | 1.44 | 1.60 | 0.36 | 688 | 1.69 | 1.68 | 0.04 | 427 | 0.70 | 0.62 | 0.31 | na | na | na | na | 585 | 0.91 | 0.92 | 0.33 |
| Feb | 671 | 1.48 | 1.52 | 0.16 | 672 | 1.42 | 1.46 | 0.23 | 584 | 1.69 | 1.67 | 0.10 | 656 | 1.66 | 1.66 | 0.06 | 414 | 0.89 | 0.90 | 0.17 | na | na | na | na | 26 | 0.42 | 0.41 | 0.13 |
| Mar | 744 | 1.49 | 1.59 | 0.31 | 694 | 1.34 | 1.34 | 0.33 | 703 | 1.55 | 1.64 | 0.24 | 708 | 1.62 | 1.61 | 0.07 | 708 | 1.09 | 1.09 | 0.16 | na | na | na | na | na | na | na | na |
| Apr | 675 | 1.42 | 1.45 | 0.60 | 718 | 1.21 | 1.21 | 0.47 | 688 | 1.31 | 1.49 | 0.49 | 677 | 1.52 | 1.52 | 0.11 | 681 | 1.06 | 1.07 | 0.04 | na | na | na | na | 50 | 0.97 | 0.98 | 0.04 |
| May | 702 | 1.21 | 1.29 | 0.50 | 722 | 1.56 | 1.60 | 0.50 | 709 | 1.13 | 1.33 | 0.54 | 534 | 1.27 | 1.28 | 0.26 | 542 | 1.07 | 1.08 | 0.05 | na | na | na | na | 84 | 0.68 | 0.63 | 0.10 |
| Jun | 712 | 1.43 | 1.58 | 0.39 | 718 | 1.46 | 1.45 | 0.25 | 689 | 1.49 | 1.56 | 0.25 | 664 | 1.41 | 1.43 | 0.12 | 680 | 1.03 | 1.03 | 0.05 | na | na | na | na | na | na | na | na |
| Jul | 732 | 1.74 | 1.72 | 0.21 | 28 | 1.47 | 1.46 | 0.05 | 666 | 1.62 | 1.59 | 0.15 | 714 | 1.41 | 1.42 | 0.10 | 693 | 1.00 | 1.00 | 0.03 | na | na | na | na | 17 | 0.82 | 0.81 | 0.03 |
| Aug | 744 | 1.72 | 1.65 | 0.21 | na | na | na | na | na | na | na | na | 725 | 1.34 | 1.37 | 0.08 | 672 | 1.02 | 1.02 | 0.05 | na | na | na | na | na | na | na | na |
| Sep | 720 | 1.43 | 1.42 | 0.06 | na | na | na | na | na | na | na | na | 711 | 1.37 | 1.37 | 0.08 | 670 | 0.99 | 0.99 | 0.06 | na | na | na | na | na | na | na | na |
| Oct | 605 | 1.36 | 1.36 | 0.02 | na | na | na | na | 586 | 1.40 | 1.44 | 0.14 | 740 | 1.45 | 1.45 | 0.06 | 662 | 0.91 | 0.91 | 0.19 | na | na | na | na | na | na | na | na |
| Nov | 646 | 1.32 | 1.33 | 0.07 | 50 | 1.14 | 1.13 | 0.05 | 660 | 1.56 | 1.56 | 0.09 | 582 | 1.54 | 1.54 | 0.06 | 586 | 0.76 | 0.76 | 0.24 | na | na | na | na | 569 | 0.67 | 0.65 | 0.30 |
| Dec | 664 | 1.29 | 1.31 | 0.10 | 670 | 1.12 | 1.12 | 0.04 | 744 | 1.57 | 1.58 | 0.08 | 737 | 1.61 | 1.60 | 0.05 | 686 | 0.79 | 0.72 | 0.33 | 240 | 0.87 | 0.78 | 0.43 | 626 | 1.00 | 0.99 | 0.45 |
| **2015** | | | | | | | | | | | | | | | | | | | | | | | | | | | | |
| Jan | na | na | na | na | na | na | na | na | 730 | 1.56 | 1.56 | 0.10 | 139 | 1.57 | 1.57 | 0.04 | 648 | 0.94 | 0.83 | 0.41 | 710 | 0.88 | 0.81 | 0.51 | 711 | 0.82 | 0.82 | 0.31 |
| Feb | na | na | na | na | na | na | na | na | 665 | 1.52 | 1.50 | 0.14 | 560 | 1.58 | 1.59 | 0.06 | 520 | 0.95 | 0.92 | 0.25 | 652 | 0.93 | 0.78 | 0.59 | 664 | 0.81 | 0.81 | 0.21 |
| Mar | na | na | na | na | na | na | na | na | 701 | 1.53 | 1.58 | 0.19 | 585 | 1.47 | 1.46 | 0.09 | na | na | na | na | 734 | 1.50 | 1.47 | 0.31 | 695 | 0.90 | 0.89 | 0.09 |
| Apr | na | na | na | na | na | na | na | na | 707 | 1.39 | 1.48 | 0.37 | 607 | 1.56 | 1.54 | 0.15 | na | na | na | na | 717 | 1.49 | 1.49 | 0.16 | 715 | 0.89 | 0.88 | 0.06 |
| May | na | na | na | na | 672 | 1.05 | 1.01 | 0.37 | 742 | 1.34 | 1.34 | 0.23 | 741 | 1.47 | 1.48 | 0.08 | na | na | na | na | 648 | 1.31 | 1.30 | 0.11 | 329 | 0.90 | 0.89 | 0.05 |
| Jun | na | na | na | na | 387 | 1.22 | 1.21 | 0.16 | 616 | 1.67 | 1.64 | 0.20 | 703 | 1.49 | 1.48 | 0.09 | na | na | na | na | 717 | 1.20 | 1.18 | 0.06 | na | na | na | na |
| Jul | na | na | na | na | na | na | na | na | 720 | 1.73 | 1.70 | 0.20 | 729 | 1.50 | 1.50 | 0.09 | na | na | na | na | 744 | 1.14 | 1.14 | 0.05 | na | na | na | na |
| Aug | na | na | na | na | na | na | na | na | 682 | 1.53 | 1.54 | 0.12 | 568 | 1.54 | 1.52 | 0.12 | na | na | na | na | 740 | 1.10 | 1.10 | 0.07 | na | na | na | na |
| Sep | na | na | na | na | na | na | na | na | 616 | 1.67 | 1.64 | 0.20 | 703 | 1.49 | 1.48 | 0.09 | na | na | na | na | 718 | 1.03 | 1.05 | 0.15 | na | na | na | na |
| Oct | na | na | na | na | na | na | na | na | 707 | 1.37 | 1.37 | 0.07 | 665 | 1.52 | 1.51 | 0.05 | 714 | 0.94 | 0.96 | 0.21 | 725 | 0.71 | 0.69 | 0.32 | na | na | na | na |
| Nov | na | na | na | na | na | na | na | na | 682 | 1.40 | 1.41 | 0.08 | 568 | 1.48 | 1.49 | 0.09 | 695 | 0.91 | 0.90 | 0.26 | 680 | 0.54 | 0.48 | 0.29 | na | na | na | na |
| Dec | na | na | na | na | na | na | na | na | 702 | 1.52 | 1.51 | 0.11 | 628 | 1.46 | 1.46 | 0.09 | 712 | 0.91 | 0.79 | 0.41 | 598 | 0.81 | 0.76 | 0.33 | na | na | na | na |

**Table 3:** Monthly-based statistics (number of hourly-averaged Hg(0) data (n), mean, median, standard deviation (SD)) of Hg(0) concentrations (in ng m$^{-3}$) at ground-based Polar sites over the 2011-2015 period. Note that 2013 data at DC refer to concentrations recorded at 210 cm above the snowpack. na: not available due to QA/QC invalidation, instrument failure, or because the QA/QC validation is currently in progress (2015 data).





| | GLEMOS | | | GEOS-Chem | | | GEM-MACH-Hg | | | ECHMERIT | | |
|---|---|---|---|---|---|---|---|---|---|---|---|---|
| | NSE | RMSE | PBIAS | NSE | RMSE | PBIAS | NSE | RMSE | PBIAS | NSE | RMSE | PBIAS |
| ALT | 0.12 | 0.29 | 4.9 | 0.32 | 0.25 | 1.3 | 0.49 | 0.22 | 4.1 | -0.27 | 0.34 | -10.0 |
| SND | -0.83 | 0.29 | -12.0 | -0.85 | 0.29 | -13.7 | -0.17 | 0.23 | -9.0 | -2.85 | 0.42 | -22.7 |
| NYA | 0.00 | 0.11 | -6.3 | -1.82 | 0.18 | -9.7 | -0.40 | 0.13 | -4.4 | -4.16 | 0.25 | -15.5 |
| AND | -2.76 | 0.20 | -8.3 | -2.50 | 0.19 | -12.2 | -0.26 | 0.12 | -4.1 | -6.24 | 0.28 | -16.7 |
| TR | -1.83 | 0.13 | 14.0 | -4.76 | 0.19 | 3.0 | -2.98 | 0.16 | 10.2 | -2.50 | 0.15 | -11.8 |
| DC | -0.28 | 0.19 | 16.2 | -1.07 | 0.25 | 7.5 | -1.08 | 0.25 | 16.3 | -0.32 | 0.20 | -6.6 |
| DDU | -6.10 | 0.24 | 25.4 | -8.15 | 0.27 | 16.9 | -4.87 | 0.22 | 16.7 | -0.85 | 0.12 | -5.1 |

**Table 4:** Goodness-of-fit statistics between monthly-averaged (year 2013) modeled and observed Hg(0) data at all ground-based sites: Nash-Sutcliffe efficiency (NSE, quantity without unit), root mean square error (RMSE, in ng/m$^3$), and percent bias (PBIAS, in %).

| | GEOS-Chem | | | | GEM-MACH-Hg | | | |
|---|---|---|---|---|---|---|---|---|
| | 2011 | 2012 | 2013 | 2014 | 2011 | 2012 | 2013 | 2014 |
| *Summer* | | | | | | | | |
| **ALT** | -23.9 | -1.9 | -15.4 | -17.1 | -12.3 | 11.1 | -9.2 | -10.0 |
| **SND** | 34.3 | -3.8 | -22.0 | 4.6 | 11.6 | 1.4 | -17.5 | 3.4 |
| **NYA** | -8.9 | -7.3 | -14.7 | -15.6 | -5.9 | -4.4 | -0.2 | -1.0 |
| **AND** | -13.2 | -10.4 | -11.9 | -14.1 | -7.2 | -6.8 | 3.2 | 3.0 |
| **TR** | -1.1 | -14.0 | -8.9 | -5.6 | 4.0 | -1.9 | 6.3 | 23.6 |
| **DC** | na | 1.7 | 15.6 | na | na | 8.7 | 35.6 | na |
| **DDU** | na | 0.1 | 0.0 | -8.3 | na | -3.4 | -1.7 | 8.4 |
| *Fall* | | | | | | | | |
| **ALT** | 9.4 | 11.7 | -9.8 | -9.5 | 13.4 | 14.7 | -3.6 | -3.0 |
| **SND** | -3.3 | -1.5 | -9.1 | 23.4 | 2.7 | -0.5 | -5.0 | 26.8 |
| **NYA** | -11.1 | -7.9 | -14.4 | -12.0 | -9.3 | -8.4 | -9.7 | -8.5 |
| **AND** | -12.6 | -11.1 | -15 | -12.1 | -13.4 | -12.5 | -13.9 | -6.5 |
| **TR** | -13.1 | -12.0 | -10.9 | -24.6 | -7.8 | -1.4 | -2.9 | -11.6 |
| **DC** | na | -31.5 | -22.6 | na | na | -18.6 | -43.4 | na |
| **DDU** | na | -9.6 | 1.1 | -19.9 | na | -3.2 | 2.1 | -4.4 |
| *Winter* | | | | | | | | |
| **ALT** | 11.8 | 18.5 | 11.7 | 3.3 | 12.8 | 19.2 | 16.2 | 8.0 |
| **SND** | 5.5 | 5.5 | 4.2 | 11.6 | 5.1 | 4.8 | 5.5 | 15.3 |
| **NYA** | 4.1 | 0.1 | -3.0 | -4.0 | 1.3 | -1.4 | -1.4 | -1.5 |
| **AND** | -7.6 | -9.0 | -8.0 | -7.6 | -10.1 | -11.1 | -7.2 | -6.7 |
| **TR** | 25.3 | 29.8 | 29.6 | 14.1 | 5.8 | 9.2 | 11.3 | 2.8 |
| **DC** | na | 79.9 | 39.3 | na | na | 48.4 | 17.8 | na |
| **DDU** | na | 38.5 | 50.4 | 49.4 | na | 15.4 | 26.9 | 40.4 |
| *Spring* | | | | | | | | |
| **ALT** | 3.2 | 27.4 | 29.7 | -21.8 | -23.0 | 9.3 | 11.8 | -24.0 |
| **SND** | 12.3 | -11.6 | -25.5 | -33.3 | 4.2 | -27.7 | -23.0 | -18.8 |
| **NYA** | -5.8 | -5.3 | -9.7 | -17.8 | -23.8 | -17.0 | -21.5 | -20.4 |
| **AND** | -11.5 | -13.8 | -12.4 | -16.7 | -9.3 | -16.0 | -5.5 | -7.6 |
| **TR** | na | -9.0 | 13.0 | -7.7 | na | 7.5 | 36.5 | 18.1 |
| **DC** | na | 32.6 | 22.9 | na | na | 48.8 | 34.5 | na |
| **DDU** | na | 3.2 | 73.6 | na | na | 31.9 | 62.8 | na |

**Table 5:** Percent bias (in %) between hourly-averaged modeled and observed Hg(0) data at all ground-based sites. Summer refers to Jun - Aug (Nov - Feb), fall to Sep - Nov (Mar - Apr), winter to Dec - Feb (May - Aug), and spring to Mar - May (Sep - Oct) at Arctic (Antarctic) sites. na: not available due to QA/QC invalidation, or instrument failure.



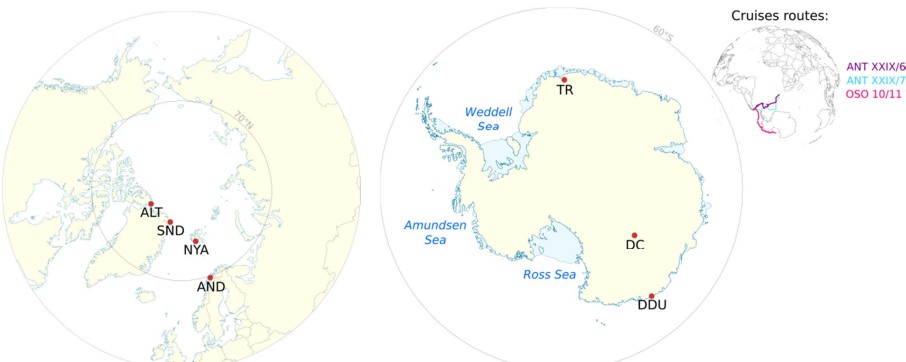

**Figure 1:** Location of the sites whose data are reported in this paper: Alert (ALT), Villum Research Station at Station Nord (SND), Zeppelin station at Ny-Ålesund (NYA), Andøya (AND), Troll (TR), Concordia Station at Dome C (DC), and Dumont d'Urville (DDU). Additionally, two cruises were performed in Antarctica: ANT XXIX/6-7 (denoted ANT in the paper) over the Weddell Sea onboard icebreaker Polarstern, and OSO 10/11 (denoted OSO in the paper) over Ross and Amundsen Seas onboard icebreaker Oden.





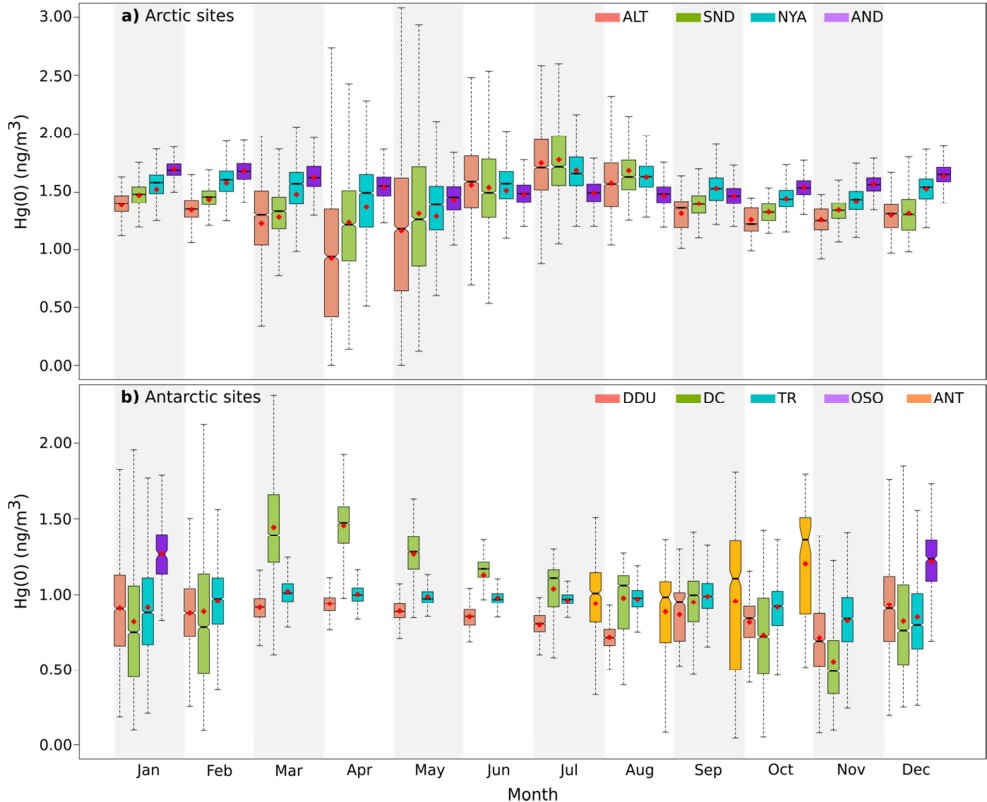

**Figure 2:** Box and whisker plots presenting the monthly Hg(0) concentration distribution at **a)** Arctic sites: ALT (red), SND (green), NYA (turquoise), AND (purple), and **b)** Antarctic sites: DDU (red), DC (green), TR (turquoise), during the OSO (purple) and ANT (orange) cruises. ♦: mean, bottom and top of the box: first and third quartiles, band inside the box: median, ends of the whiskers: lowest (highest) datum still within the 1.5 interquartile range of the lowest (upper) quartile. Outliers are not represented.





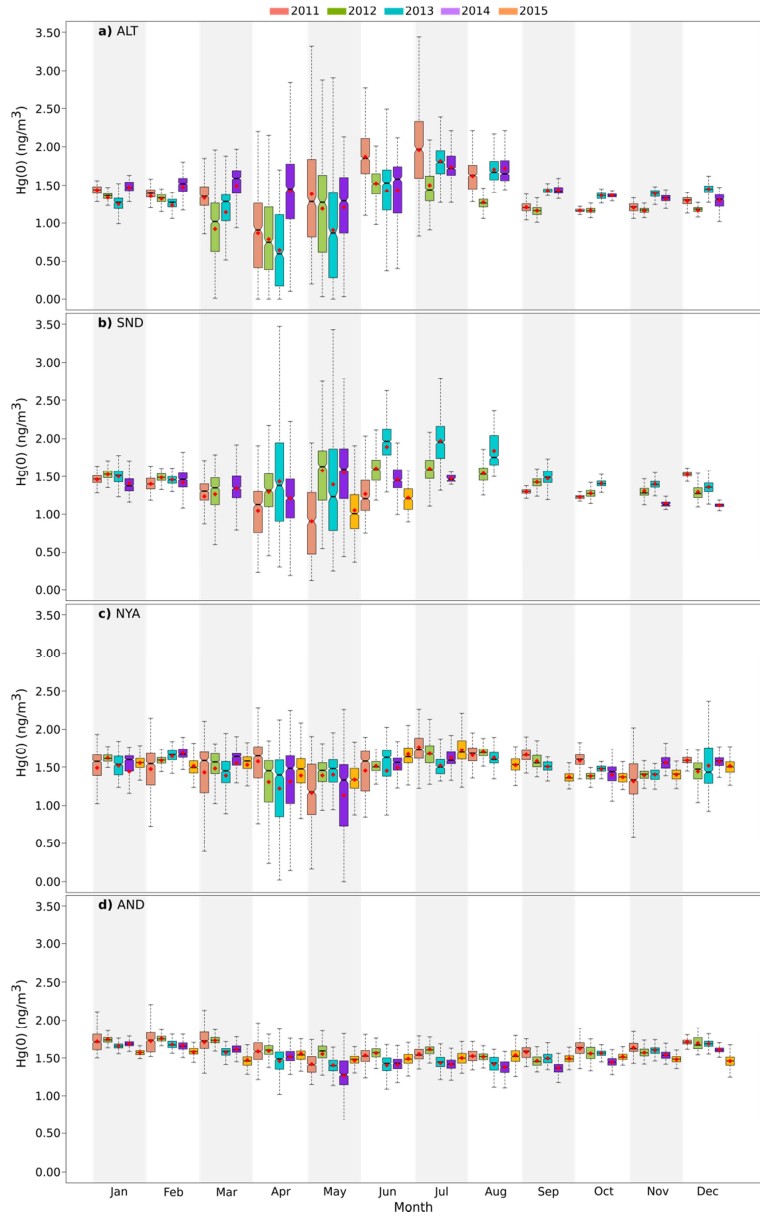

**Figure 3:** Box and whisker plots presenting the monthly Hg(0) concentration distribution at Arctic sites **a)** ALT, **b)** SND, **c)** NYA, and **d)** AND in 2011 (pink), 2012 (green), 2013 (turquoise), 2014 (purple), and 2015 (orange). ◆: mean, bottom and top of the box: first and third quartiles, band inside the box: median, ends of the whiskers: lowest (highest) datum still within the 1.5 interquartile range of the lowest (upper) quartile. Outliers are not represented.





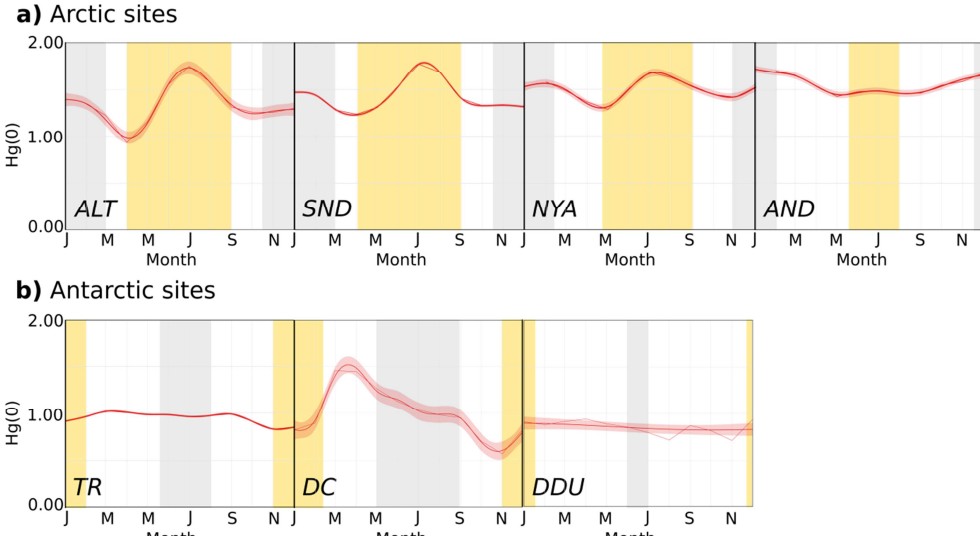

**Figure 4:** Seasonal variation (monthly mean along with the 95% confidence interval for the mean) of Hg(0) concentrations (in ng m$^{-3}$) at **a)** Arctic and **b)** Antarctic ground-based sites. Periods highlighted in yellow refer to 24-h sunlight and periods highlighted in grey to 24-h darkness. Summer refers to June – August (November – February), fall to September – November (March – April), winter to December – February (May – August), and spring to March – May (September – October) at Arctic (Antarctic) sites.





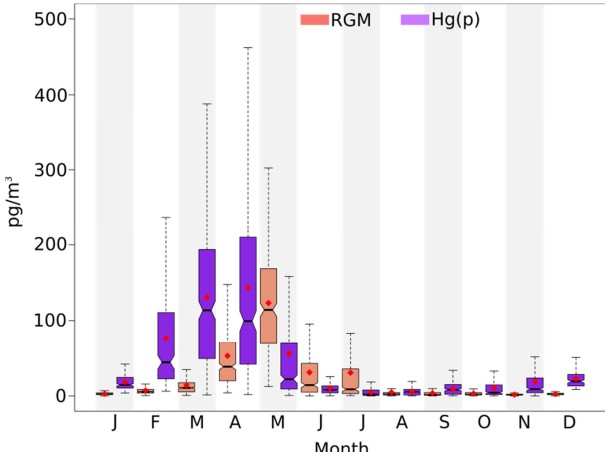

**Figure 5:** Box and whisker plots presenting the monthly RGM (in red) and Hg(p) (in violet) concentration distribution (in pg m$^{-3}$) at ALT over the 2011-2014 period. ◆: mean, bottom and top of the box: first and third quartiles, band inside the box: median, ends of the whiskers: lowest (highest) datum still within the 1.5 interquartile range of the lowest (upper) quartile. Outliers are not represented.




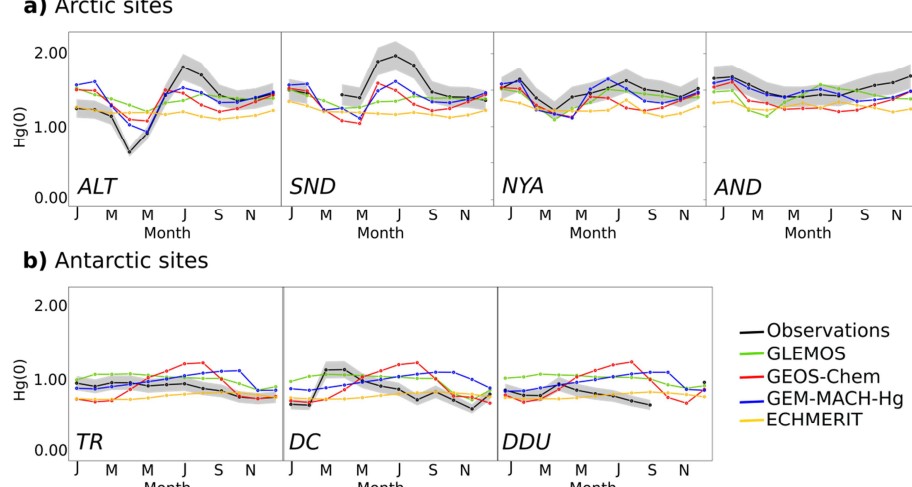

**Figure 6:** Year 2013 monthly-averaged Hg(0) concentrations (in ng m$^{-3}$) at **a)** Arctic and **b)** Antarctic ground-based sites: observations (in black) and concentrations according to the four global models (GLEMOS in green, GEOS-Chem in blue, GEM-MACH-Hg in red, ECHMERIT in yellow). The gray shaded regions indicate a 10 % uncertainty for observations.





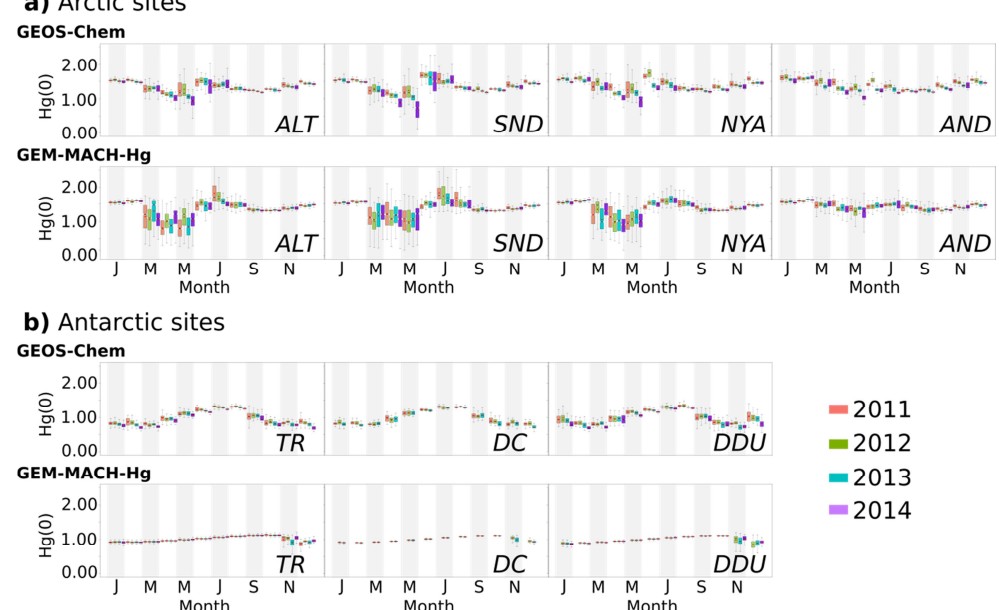

**Figure 7:** Box and whisker plots presenting the monthly Hg(0) concentration distribution at **a)** Arctic and **b)** Antarctic ground-based sites as simulated by GEOS-Chem and GEM-MACH-Hg in 2011 (pink), 2012 (green), 2013 (turquoise), and 2014 (purple). ◆: mean, bottom and top of the box: first and third quartiles, band inside the box: median, ends of the whiskers: lowest (highest) datum still within the 1.5 interquartile range of the lowest (upper) quartile. Outliers are not represented.





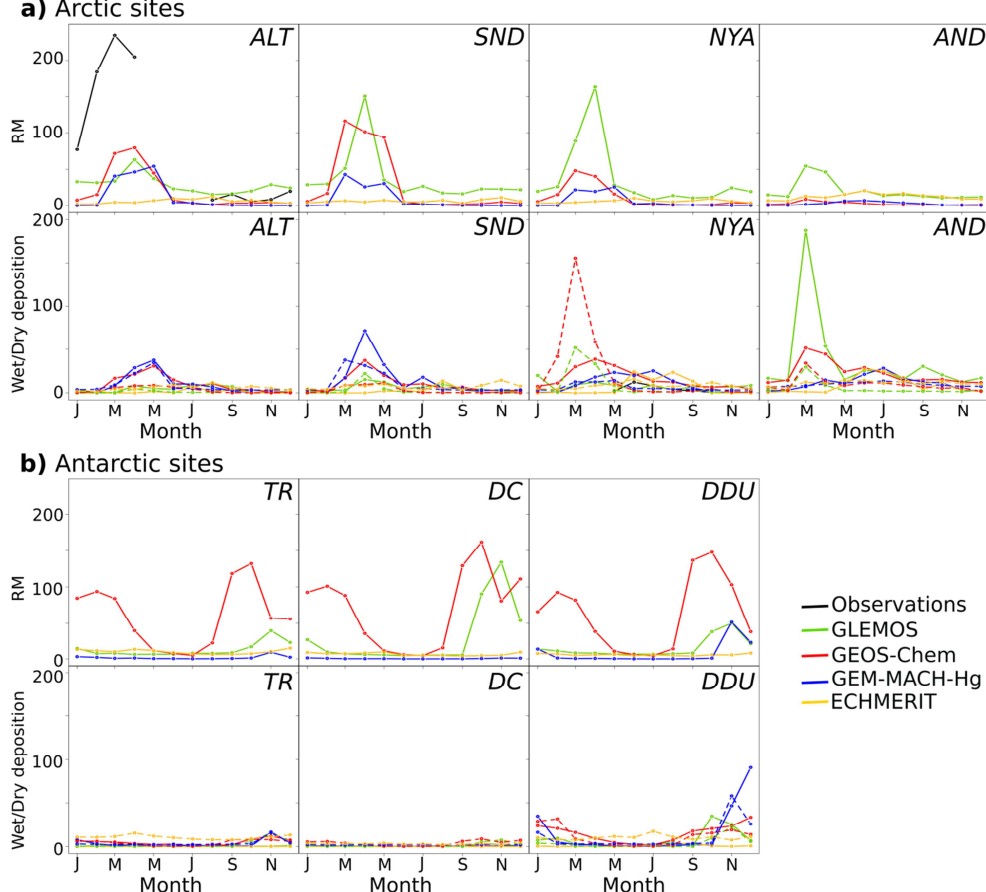

**Figure 8:** Year 2013 monthly-averaged mean reactive mercury (RM) concentrations (in pg m$^{-3}$) along with mean wet (solid line) and dry (dashed line) deposition (in ng m$^{-2}$ day$^{-1}$) at **a)** Arctic and **b)** Antarctic ground-based sites: observations (in black) and concentrations according to the four global models (GLEMOS in green, GEOS-Chem in red, GEM-MACH-Hg in blue, ECHMERIT in yellow). Note that RM (wet deposition) observations are available at ALT (NYA) only.





**Figure 9:** Box and whisker plots presenting the monthly Hg(0) concentration distribution at ground-based Antarctic sites **a)** TR, **b)** DC, and **c)** DDU in 2011 (pink), 2012 (green), 2013 (turquoise), 2014 (purple), and 2015 (orange). ◆: mean, bottom and top of the box: first and third quartiles, band inside the box: median, ends of the whiskers: lowest (highest) datum still within the 1.5 interquartile range of the lowest (upper) quartile. Outliers are not represented.