# Peer review of "Chemical cycling and deposition of atmospheric mercury"

_Atmospheric Chemistry and Physics, 2016_

## Referee Comment (RC1) · L.-E. Heimburger (Referee) · 18 Jul 2016

Interesting read, nicely resuming the last 4 years of atmospheric mercury research in polar regions. I only have a few minor comments/edits: L223ff: how does this compare to Sommar, J., M. E. Andersson and H. W. Jacobi (2010). "Circumpolar measurements of speciated mercury, ozone and carbon monoxide in the boundary layer of the Arctic Ocean." Atmos. Chem. Phys. 10(11): 5031-5045, and Yu, J., Z. Xie, H. Kang, Z. Li, C. Sun, L. Bian and P. Zhang (2014). "High variability of atmospheric mercury in the summertime boundary layer through the central Arctic Ocean." Sci. Rep. 4. The latter 2012 is also missing in the figure 1 of current arctic data, and could be discussed a bit more. Is there no other ship-based arctic data? L244: better unify units to °C as used

above l226 L359: 38 and 38% sounds odd L366: remove frequency L499: remove extra period in GEO-C hem L675: what is the threshold for Antarctic AMDEs? You mentioned 1ng m3 for Arctic AMDEs before?
* * *

---

## Referee Comment (RC2) · Anonymous Referee #2 · 25 Jul 2016

A few minor suggestions:

line 568: interesting that the models underestimate RGM since the KCl denuder collection method is thought to collect RGM with <100% efficiency. I did not immediately see that a reason for the discrepancy was given. Similar to results shown in Weiss-Penzias et al ACP 2015, Figure 5, where the GEOS-Chem model underpredicted a high RGM event at Desert Research Institute site in Nevada USA.

line 938: passive samplers are mentioned in too casual a way as a possible solution to obtaining year round RGM data. Have they been adequately tested to know their collection efficiencies and potential biases? This is mentioned in point number 2. Maybe combine points 1 and 2?

[Figure]

line 949: from how many sites in polar regions would snow samples need to be taken in order to have a better understanding of Hg wet and dry deposition?

---

## Author Comment (AC1) · 12 Aug 2016

**"Chemical cycling and deposition of atmospheric mercury in Polar Regions: review of recent measurements and comparison with models" by H. Angot et al.**

**Response to referee comments by Referee #1.**

We provide below a point-by-point reply to the comments (points raised by the referee in bold, changes made in the manuscript in red).

**Interesting read, nicely resuming the last 4 years of atmospheric mercury research in Polar Regions. I only have a few minor comments/edits:**

**- L223ff: how does this compare to *Sommar, J., M. E. Andersson and H. W. Jacobi (2010). "Circumpolar measurements of speciated mercury, ozone and carbon monoxide in the boundary layer of the Arctic Ocean." Atmos. Chem. Phys. 10(11): 5031-5045* and *Yu, J., Z. Xie, H. Kang, Z. Li, C. Sun, L. Bian and P. Zhang (2014). "High variability of atmospheric mercury in the summertime boundary layer through the central Arctic Ocean." Sci. Rep. 4.*? The latter 2012 is also missing in the figure 1 of current arctic data, and could be discussed a bit more. Is there no other ship-based arctic data?**

The path of the CHINARE 2012 cruise (Yu et al., 2014) has been added in Figure 1 of the revised manuscript (see below).

[Figure]

**Figure 1:** Location of a) Arctic and b) Antarctic ground-based sites whose data are reported in this paper: Alert (ALT), Villum Research Station at Station Nord (SND), Zeppelin station at Ny-Ålesund (NYA), Andøya (AND), Troll (TR), Concordia Station at Dome C (DC), and Dumont d'Urville (DDU). Additionally, the approximate path of cruises performed in recent years (2011-2015) is given: CHINARE 2012 in the Arctic onboard the Chinese vessel Xuelong (in blue), ANT XXIX/6-7 (denoted ANT in the paper) over the Weddell Sea onboard icebreaker Polarstern (in yellow and purple), and OSO 10/11 (denoted OSO in the paper) over Ross and Amundsen Seas onboard icebreaker Oden (in orange).

Additionally, results from the two aforementioned cruises are discussed here and there in the revised manuscript (where appropriate):

Lines 320-322:
"In contrast, lower concentrations were found in the Chukchi Sea in July ($1.17 \pm 0.38$ ng m$^{-3}$) than in September ($1.51 \pm 0.79$ ng m$^{-3}$) during the CHINARE 2012 expedition (Yu et al., 2014)."

Lines 432-435:
"Inhomogeneous distributions of Hg(0) were observed over the Arctic Ocean during the CHINARE 2012 (Yu et al., 2014) and the Beringia 2005 (Sommar et al., 2010) expeditions. Both studies reported a rapid increase of concentrations in air when entering the ice-covered waters, highlighting the influence of sea ice dynamics on Hg(0) concentrations."

**- L244: better unify units to °C as used above L226.**

This has been corrected in the revised manuscript.

**- L359: 38 and 38 % sounds odd.**

This has been reworded in the revised manuscript:

"Over the 2011-2015 period, AMDEs at NYA are evenly distributed between April and May (38 % of the time in both cases) , and fewer in March and June (14 and 10 % of the time, respectively)."

**- L366: remove frequency.**

This has been removed in the revised manuscript.

**- L499: remove extra period in GEOS-C hem.**

Done.

**- L675: what is the threshold for Antarctic AMDEs? You mentioned 1 ng m$^{-3}$ for Arctic AMDEs before?**

This is specified line 672:

"AMDEs in Antarctica are operationally defined as Hg(0) concentrations below 0.60 ng m$^{-3}$ (Pfaffhuber et al., 2012)".

References:

Pfaffhuber, K. A., Berg, T., Hirdman, D., and Stohl, A.: Atmospheric mercury observations from Antarctica: seasonal variation and source and sink region calculations, Atmospheric Chemistry and Physics, 12, 3241-3251, 2012.

Yu, J., Xie, Z., Kang, H., Li, Z., Sun, C., Bian, L., and Zhang, P.: High variability of atmospheric mercury in the summertime boundary layer through the central Arctic Ocean, Scientific Reports, 4, 6091, 10.1038/srep06091 http://www.nature.com/articles/srep06091#supplementary-information, 2014.

---

## Author Comment (AC2) · 12 Aug 2016

**"Chemical cycling and deposition of atmospheric mercury in Polar Regions: review of recent measurements and comparison with models" by H. Angot et al.**

**Response to referee comments by Referee #2.**

We provide below a point-by-point reply to the comments (points raised by the referee in bold, changes made in the manuscript in red).

**A few minor suggestions:**

**-line 568: interesting that the models underestimate RGM since the KCl denuder collection method is thought to collect RGM with < 100 % efficiency. I did not immediately see that a reason for the discrepancy was given. Similar to the results shown in Weiss-Penzias et al ACP 2015, Figure 5, where the GEOS-Chem model underpredicted high RGM event at Desert Research Institute site in Nevada USA.**

Indeed, several studies highlighted the inefficient collection of Hg(II) with a KCl-coated denuder leading to an underestimation of RM concentrations by a factor of 1.3-3.7 (Huang et al., 2013). This suggests that the underestimation of RM concentrations by current models might be even greater.

**-line 938: passive samplers are mentioned in too casual a way as a possible solution to obtaining year round RGM data. Have they been adequately tested to know their collection efficiencies and potential biases? This is mentioned in point number 2. Maybe combine points 1 and 2?**

We agree that passive samplers have to be adequately tested first. This has been corrected in the revised manuscript:
"Passive samplers, such as Polyethersulfone cation exchange membranes, could provide an alternative (Huang et al., 2014) but further tests are needed to assess their collection efficiency and potential biases".

Point number 1 deals with RM measurements while point number 2 deals with Hg(II) speciation. We would rather not combine them. Point number 2 has been slightly modified in the revised manuscript to avoid redundancy:
"Recent advancement on analytical techniques may offer new insights into Hg(II) speciation (Huang et al., 2013;  Jones et al., 2016) but  further research is still needed ."

**-line 949: from how many sites in Polar Regions would snow samples need to be taken in order to have a better understanding of Hg wet and dry deposition?**

We believe that collecting surface snow samples at all sites carrying out long-term atmospheric Hg monitoring would be a good start.

References:

Huang, J., Miller, M. B., Weiss-Penzias, P., and Gustin, M. S.: Comparison of gaseous oxidized mercury measured by KCl-coated denuders, and nylon and cation exchange membranes, Environmental Science and Technology, 47, 7307-7316, 2013.

Huang, J., Lyman, S. N., Hartman, J. S., and Gustin, M. S.: A review of passive sampling systems for ambient air mercury measurements, Environmental Science: Processes & Impacts, 16, 374-392, 10.1039/C3EM00501A, 2014.

Jones, C. P., Lyman, S. N., Jaffe, D. A., Allen, T., and O'Neil, T. L.: Detection and quantification of gas-phase oxidized mercury compounds by GC/MS, Atmos. Meas. Tech., 9, 2195-2205, 10.5194/amt-9-2195-2016, 2016.